# DOMINO: DISCOVERING SYSTEMATIC ERRORS WITH CROSS-MODAL EMBEDDINGS

**Sabri Eyuboglu**\*, **Maya Varma**\*, **Khaled Saab**\*, **Jean-Benoit Delbrouck, Christopher Lee-Messer, Jared Dunnmon, James Zou, Christopher Ré**
Stanford University, USA; {eyuboglu,mvarma2,ksaab}@stanford.edu; (\*Equal contribution)

## ABSTRACT

Machine learning models that achieve high overall accuracy often make systematic errors on important subsets (or *slices*) of data. Identifying underperforming slices is particularly challenging when working with high-dimensional inputs (*e.g.* images, audio), where important slices are often unlabeled. In order to address this issue, recent studies have proposed automated slice discovery methods (SDMs), which leverage learned model representations to mine input data for slices on which a model performs poorly. To be useful to a practitioner, these methods must identify slices that are both underperforming and *coherent* (*i.e.* united by a human-understandable concept). However, no quantitative evaluation framework currently exists for rigorously assessing SDMs with respect to these criteria. Additionally, prior qualitative evaluations have shown that SDMs often identify slices that are incoherent. In this work, we address these challenges by first designing a principled evaluation framework that enables a quantitative comparison of SDMs across 1,235 slice discovery settings in three input domains (natural images, medical images, and time-series data). Then, motivated by the recent development of powerful cross-modal representation learning approaches, we present *Domino*, an SDM that leverages cross-modal embeddings and a novel error-aware mixture model to discover and describe coherent slices. We find that Domino accurately identifies 36% of the 1,235 slices in our framework – a 12 percentage point improvement over prior methods. Further, Domino is the first SDM that can provide natural language descriptions of identified slices, correctly generating the exact name of the slice in 35% of settings.

## 1 INTRODUCTION

Machine learning models often make systematic errors on important subsets (or *slices*) of data[1]. For instance, models trained to detect collapsed lungs in chest X-rays have been shown to make predictions based on the presence of chest drains, a device typically used during treatment (Oakden-Rayner et al., 2019). As a result, these models frequently make prediction errors on cases without chest drains, a critical data slice where false negative predictions could be life-threatening. Similar performance gaps across slices have been observed in radiograph classification (Badgeley et al., 2019; Zech et al., 2018; DeGrave et al., 2021), melanoma detection (Winkler et al., 2019), natural language processing (Orr et al., 2020; Goel et al., 2021), and object detection (de Vries et al., 2019), among others. If underperforming slices can be accurately identified and labeled, we can then improve model robustness by either updating the training dataset or using robust optimization techniques (Zhang et al., 2018; Sagawa et al., 2020).

However, identifying underperforming slices is difficult in practice. When working with high-dimensional inputs (*e.g.* images, time-series data, video), slices are often "hidden", meaning that they cannot easily be extracted from the inputs and are not annotated in metadata (Oakden-Rayner et al., 2019). For instance, in the collapsed lung example above, the absence of chest drains is challenging to identify from raw image data and may not be explicitly labeled in metadata. In this setting, we must perform *slice discovery*: the task of mining unstructured input data for semantically meaningful subgroups on which the model performs poorly.

---

[1]We define a data *slice* as a group of data examples that share an attribute or characteristic.

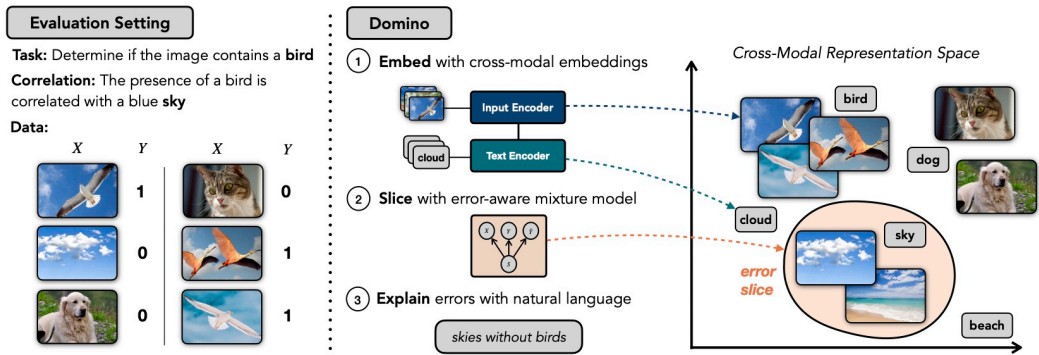

Figure 1: **Proposed Approach**. (Left) We design an evaluation framework to systematically compare SDMs across diverse slice settings. Here, the example slice setting includes a dataset that displays a strong correlation between the presence of birds and skies. (Right) A classifier trained to detect the presence of birds makes false positive predictions on skies without birds. We present *Domino*, a novel SDM that uses cross-modal embeddings to identify and describe the error slice.

In modern machine learning workflows, practitioners commonly attempt slice discovery with a combination of feature-based interpretability methods (*e.g.* GradCAM, LIME) and manual inspection (Selvaraju et al., 2017; Ribeiro et al., 2016). However, these approaches are time-consuming and susceptible to confirmation bias (Adebayo et al., 2018). As a result, recent works have proposed automated *slice discovery methods* (SDMs), which use learned input representations to identify semantically meaningful slices where the model makes prediction errors (d'Eon et al., 2021; Yeh et al., 2020a; Sohoni et al., 2020; Kim et al., 2018; Singla et al., 2021). An ideal SDM should automatically identify data slices that fulfill two desiderata: (a) slices should contain examples on which the model *underperforms*, or has a high error rate and (b) slices should contain examples that are *coherent*, or align closely with a human-understandable concept. An SDM that is able to reliably satisfy these desiderata across a wide range of settings has yet to be demonstrated for two reasons:

**Issue 1:** *No quantitative evaluation framework exists for measuring performance of SDMs with respect to these desiderata.* Existing SDM evaluations are either qualitative (d'Eon et al., 2021), performed on purely synthetic data (Yeh et al., 2020a), or consider only a small selection of tasks and slices (Sohoni et al., 2020). A comprehensive evaluation framework should be quantitative, use realistic data, cover a broad range of contexts, and evaluate both underperformance and coherence. Currently, no datasets or frameworks exist to support such an evaluation, making it difficult to evaluate the tradeoffs among prior SDMs.

**Issue 2:** *Prior qualitative evaluations have demonstrated that existing SDMs often identify slices that are incoherent.* A practically useful SDM should discover coherent slices that are understandable by a domain expert. For example, in the chest X-ray setting described earlier, the slice "patients without chest drains" is meaningful to a physician. Slice coherence has previously been evaluated qualitatively by requiring users to manually inspect examples and identify common attributes (d'Eon et al., 2021; Yeh et al., 2020a). Such evaluations have shown that discovered slices often do not align with concepts understandable to a domain expert. Additionally, even if slices do align well with concepts, it may be difficult for humans to identify the shared attribute. Thus, an ideal SDM would not only output coherent slices, but also identify the concept connecting examples in each slice.

In this work, we address both of these issues by (1) developing a framework to quantitatively evaluate the effectiveness of slice discovery methods at scale and (2) leveraging this framework to demonstrate that a powerful class of recently-developed cross-modal embeddings can be used to create an SDM that satisfies the above desiderata. Our approach – *Domino* – identifies coherent slices and generates automated slice descriptions.

After formally describing the slice discovery problem in Section 2, we introduce an evaluation framework for rigorously assessing SDM performance in Section 3. We curate a set of 1,235 slice discovery settings, each consisting of a real-world dataset, a trained model, and one or more "ground truth" slices corresponding to a concept in the domain. During evaluation, the SDM is provided with the dataset and the model, and we measure if the labeled slices can be successfully identified. We find that existing methods identify "ground truth" slices in no more than 23% of these settings.

Motivated by the recent development of large cross-modal representation learning approaches (*e.g.* CLIP) that embed inputs and text in the same latent representation space, in Section 4 we present *Domino*, a novel SDM that uses cross-modal embeddings to identify coherent slices. Cross-modal representations incorporate semantic meaning from text into input embeddings, which we demonstrate can improve slice coherence and enable the generation of slice descriptions. *Domino* embeds inputs alongside natural language with cross-modal representations, identifies coherent slices with an error-aware Gaussian mixture model, and generates natural language descriptions for discovered slices. In Section 5, we use our evaluation framework to show that Domino identifies 36% of the "ground truth" coherent slices across three input domains (natural images, medical images, and time-series) – a 12 percentage point improvement over existing methods.

## 2 RELATED WORK

**Slice performance gaps.** Machine learning models are often limited by the presence of underperforming slices. In Section A.2, we provide a survey of underperforming slices reported by prior studies across a range of application domains and slice types. Oakden-Rayner et al. (2019) referred to this problem as "hidden stratification" and motivated the need for slice discovery techniques.

**Slice discovery.** Prior work on slice discovery has predominantly focused on input datasets with rich structure (*e.g.* tabular data) or metadata, where slicing can generally be performed with predicates (*e.g.* nationality = Peruvian) (Chung et al., 2019; Sagadeeva & Boehm, 2021; Chen et al., 2021). The slice discovery problem becomes particularly complex when input data lacks explicit structure (*e.g.* images, audio, etc.) and metadata, and three recent studies present methods for performing slice discovery in this unstructured setting (d'Eon et al., 2021; Sohoni et al., 2020; Kim et al., 2018; Singla et al., 2021). The proposed SDMs follow two steps: (1) embed input data in a representation space and (2) identify underperforming slices using clustering or dimensionality reduction techniques. These SDMs are typically evaluated by measuring performance over a limited number of slice settings or by performing qualitative assessments. The trade-offs between these SDMs have not been systematically compared, and as a result, the conditions under which these SDMs succeed at identifying coherent slices remain unclear.

**Benchmark datasets for machine learning robustness.** Recently, several benchmark datasets have been proposed for evaluating the performance of models on dataset shifts. These benchmarks are valuable because they provide labels specifying important slices of data. However, these datasets do not suffice for systematic evaluations of SDMs because they either only annotate a small number of slices (Koh et al., 2021) or do not provide pretrained models that are known to underperform on the slices (He et al., 2021; Khosla et al., 2011; Hendrycks & Dietterich, 2019; Liang & Zou, 2021).

**Cross-modal embeddings.** Cross-modal representation learning approaches, which embed input data and text in the same representation space, yield powerful embeddings that have contributed to large performance improvements across information retrieval and classification tasks. Cross-modal models generate semantically meaningful input representations that have been shown to be highly effective on zero-shot classification tasks (Radford et al., 2021). Cross-modal models that have inspired our work include CLIP for natural images (Radford et al., 2021), ConVIRT for medical images (Zhang et al., 2020), and WikiSatNet (Uzkent et al., 2019) for satellite imagery.

## 3 SLICE DISCOVERY PRELIMINARIES

In this section, we formulate the slice discovery problem. Consider a standard classification setting with input $X \in \mathcal{X}$ (*e.g.* an image, time-series, or graph) and label $Y \in \mathcal{Y}$ over $|\mathcal{Y}|$ classes. Additionally, assume there exists a set of $k$ *slices* that partition the data into *coherent* (potentially overlapping) subgroups, where each subgroup captures a distinct concept or attribute that would be familiar to a domain expert. For each input, we represent slice membership as $\mathbf{S} = \{S^{(j)}\}_{j=1}^{k} \in \{0,1\}^k$. As an example, in the scenario presented in Section 1, $X$ represents chest X-rays, $Y$ is a binary label indicating the presence of collapsed lungs, and $\mathbf{S} = \{S^{(1)}, S^{(2)}\}$ represents two slices: one consisting of normal X-rays *with* chest drains and the other consisting of collapsed lungs *without* chest drains.

The inputs, labels, and slices vary jointly according to a probability distribution $P(X, Y, \mathbf{S})$ over $\mathcal{X} \times \mathcal{Y} \times \{0,1\}^k$. We assume that training, validation and test data are drawn independently and

identically from this distribution. For some application-dependent value of $\epsilon$, a model $h_\theta : \mathcal{X} \to \mathcal{Y}$ exhibits degraded performance with respect to a slice $S^{(j)}$ and metric $\ell : \mathcal{Y} \times \mathcal{Y} \to \mathbb{R}$ if $\mathbb{E}_{X,Y|S^{(j)}=1}[\ell(h_\theta(X), Y)] < \mathbb{E}_{X,Y|S^{(j)}=0}[\ell(h_\theta(X), Y)] - \epsilon$.

Assuming that a trained classifier $h_\theta : \mathcal{X} \to \mathcal{Y}$ exhibits degraded performance on each of the $k$ slices in $\mathbf{S}$, we define the *slice discovery problem* as follows:

- **Inputs**: a trained classifier $h_\theta$ and labeled dataset $\mathcal{D} = \{(x_i, y_i)\}_{i=1}^n$ with $n$ samples drawn from $P(X, Y)$.

- **Output**: a set of $\hat{k}$ slicing functions $\Psi = \{\psi^{(j)} : \mathcal{X} \times \mathcal{Y} \to \{0, 1\}\}_{j=1}^{\hat{k}}$ that partition the data into $\hat{k}$ subgroups.

We consider an output to be successful if, for each ground truth slice $S^{(u)}$, a slicing function $\psi^{(v)}$ predicts $S^{(u)}$ with precision above some threshold $\beta$:

$$\forall u \in [k]. \quad \exists v \in [\hat{k}]. \quad P(S^{(u)} = 1 | \psi^{(v)}(X, Y) = 1) > \beta.$$

A *slice discovery method* (SDM), $M(\mathcal{D}, h_\theta) \to \Psi$, aims to solve the slice discovery problem.

## 4 SLICE DISCOVERY EVALUATION FRAMEWORK

It is challenging to measure how well an SDM satisfies the following desiderata outlined in Section 1: (a) the model $h_\theta$ should exhibit degraded performance on slices predicted by the SDM and (b) slices predicted by the SDM should be coherent. Most publicly available machine learning datasets do not provide labels identifying coherent, underperforming slices. As a result, existing evaluations are either qualitative (d'Eon et al., 2021), synthetic (Yeh et al., 2020b), or small-scale (Sohoni et al., 2020). In this section, we propose an evaluation framework for estimating the success rate of an SDM: how often it successfully identifies the coherent slices on which the model underperforms.

We propose evaluating SDMs across large sets of slice discovery *settings*, each consisting of: (1) a labeled **dataset** $\mathcal{D} = \{(x_i, y_i)\}_{i=1}^n$, (2) a **model** $h_\theta$ trained on $\mathcal{D}$, and (3) ground truth **slice annotations** $\{\mathbf{s}_i\}_{i=1}^n$ for one or more coherent slices $\mathbf{S}$ on which the model $h_\theta$ exhibits degraded performance. As discussed in Section 3, (1) and (2) correspond to the inputs to the SDM, while (3) corresponds to the expected output. Using these slice discovery settings, we can estimate how often an SDM $M(\mathcal{D}, f_\theta) \to \Psi$ successfully identifies the slices $\mathbf{S}$. Algorithm 1 details the procedure.

---

**Algorithm 1** SDM Evaluation Process

**for** $(\mathcal{D}, \mathbf{s}, h_\theta) \in$ settings **do**
    $\Psi \leftarrow M(\mathcal{D}_{\text{valid}}, h_\theta)$ ▷ Fit the SDM on the validation set, yielding a set of slicing functions $\Psi$
    **for** $j \in [\hat{k}]$ **do**
        $\hat{\mathbf{s}}^{(j)} \leftarrow \psi^{(j)}(\mathcal{D}_{\text{test}})$     ▷ Apply the slicing functions to the test set, yielding $\hat{\mathbf{s}} \in [0, 1]^{n_{\text{test}}}$.
    **end for**
    metrics $\leftarrow \{\max_{j \in [\hat{k}]} L(\mathbf{s}^{(i)}, \hat{\mathbf{s}}^{(j)})\}_{i=1}^k$     ▷ Compute metric $L$ comparing $\hat{\mathbf{s}}$ and $\mathbf{s}$
**end for**

---

In Section 4.1, we propose a process for generating representative slice discovery settings and in Section 4.2, we describe the evaluation metrics $L$ and datasets that we use.

### 4.1 GENERATING SLICE DISCOVERY SETTINGS

To obtain accurate estimates of SDM performance, our slice discovery settings should be both representative of real-world slice discovery settings and large in number. However, such slice discovery settings aren't readily available, since few datasets specify slices on which models underperform.

Here, we propose a framework for programmatically generating a large number of realistic slice discovery settings. We begin with a base dataset $\mathcal{D}_{\text{base}}$ that has either a hierarchical label structure (*e.g.* ImageNet) or rich metadata accompanying each example (*e.g.*. CelebA). We select a target

variable $Y$ and slice variable $\mathbf{S}$, each defined in terms of the class structure or metadata. This allows us to derive target and slice labels $\{(y_i, \mathbf{s}_i)\}_{i=1}^n$ directly from the dataset. In addition, because the slice $\mathbf{S}$ is defined in terms of meaningful annotations, we know that the slice is coherent. After selecting a target variable and slice variable, we (1) generate a dataset $\mathcal{D}$ (Section 4.1.1) and (2) generate a model $h_\theta$ that exhibits degraded performance with respect to $\mathbf{S}$ (Section 4.1.2).

### 4.1.1 DATASET GENERATION

We categorize each slice discovery setting based on the underlying reason that the model $h_\theta$ exhibits degraded performance on the slices $\mathbf{S}$. We survey the literature for examples of underperforming slices in the wild, which we document in Section A.2. Based on our survey and prior work (Oakden-Rayner et al., 2019), we identify three common slice types: *rare* slices, *correlation* slices, and *noisy label* slices. We provide expanded descriptions in Section A.3.

**Rare slice.** Models often underperform on rare data slices, which consist of data subclasses that occur infrequently in the training set (*e.g.* patients with a rare diseases, photos taken at night). To generate settings with rare slices, we construct $\mathcal{D}$ such that for a given class label $Y$, elements in subclass $C$ occur with proportion $\alpha$, where $0.01 < \alpha < 0.1$.

**Correlation slice.** If a target variable $Y$ (*e.g.* pneumothorax) is correlated with another variable $C$ (*e.g.* chest tubes), the model may learn to rely on $C$ to make predictions. To generate settings with correlation slices, we construct $\mathcal{D}$ such that a linear correlation of strength $\alpha$ exists between the target variable and other class labels, where $0.2 < \alpha < 0.8$. (Details in Section A.3.2).

**Noisy label slice.** Particular slices of data may exhibit higher label error rates than the rest of the training distribution. To generate settings with noisy labels, we construct $\mathcal{D}$ such that for each class label $Y$, the elements in subclass $C$ exhibit label noise with probability $\alpha$, where $0.01 < \alpha < 0.3$.

### 4.1.2 MODEL GENERATION

Each slice discovery setting includes a model $h_\theta$ that exhibits degraded performance with respect to a set of slices $\mathbf{S}$. We propose generating two classes of models: (a) trained models, which are realistic, and (b) synthetic models, which provide greater control over the evaluation.

**Trained Models.** We train a distinct model $h_\theta$ across each of our generated datasets $D$. If the model $h_\theta$ exhibits a degradation in performance with respect to the slices $\mathbf{S}$, then the model $h_\theta$, dataset $D$, and slices $\mathbf{S}$ will comprise a valid slice discovery setting. (See Section 3 for the formal definition of degraded performance; we use accuracy for our metric $\ell$ and set $\epsilon = 10$.)

**Synthetic Models.** Because the trained model $h_\theta$ may underperform on coherent slices other than $\mathbf{S}$, an SDM that fails to identify $\mathbf{S}$ may still recover other coherent, underperforming slices. This complicates the interpretation of our results. To address this subtle issue, we also create settings using synthetic models $\bar{h} : [0,1]^k \times \mathcal{Y} \to [0,1]$ that simulate predictions. This allows us to distribute errors outside of $\mathbf{S}$ randomly, ensuring that there are likely no underperforming, coherent slices outside of $S$. We simulate predictions by sampling from beta distributions (see Section A.3.4).

## 4.2 EVALUATION APPROACH

We instantiate our evaluation framework by generating 1,235 slice discovery settings across a number of different tasks, applications, and base datasets. Detailed statistics on our settings are provided in Table 1. We generate slice discovery settings in the following three domains:

**Natural Images (CelebA and ImageNet):** The CelebFaces Attributes Dataset (CelebA) includes over 200k images with 40 labeled attributes (Liu et al., 2015). ImageNet includes 1.2 million images across 1000 labeled classes organized in a hierarchical structure (Deng et al., 2009; Fellbaum, 1998).

**Medical Images (MIMIC-CXR):** The MIMIC Chest X-Ray (MIMIC-CXR) dataset includes 377,110 chest x-rays collected from the Beth Israel Deaconess Medical Center. Annotations indicate the presence or absence of fourteen conditions (Johnson et al., 2019; 2020).

**Medical Time-Series Data (EEG):** In addition to our image modalities, we also explore time-series data. We obtain a dataset of short 12 second electroencephalography (EEG) signals, which have been used in prior work for predicting the onset of seizures (Saab et al., 2020).

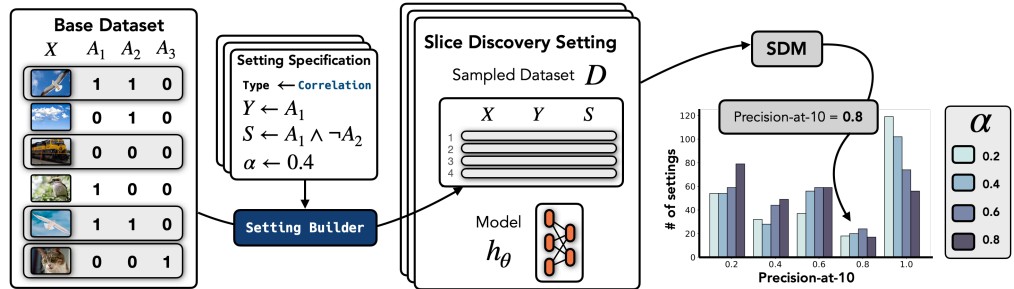

Figure 2: **Evaluation Framework**. We propose a framework for generating slice discovery settings from any base dataset with class structure or metadata.

We evaluate SDM performance with *precision-at-k*, which measures the proportion of the top $k$ elements in the discovered slice that are in the ground truth slice. We use $k = 10$ in this work.

## 5 DOMINO

In this section, we introduce *Domino*, an SDM that uses cross-modal embeddings to identify coherent slices and generate natural language descriptions. Domino follows a three-step procedure:

1. **Embed** (Section 5.1): We encode the inputs $\{x_i\}_{i=1}^n$ in a cross-modal embedding space via a function $g_{\text{input}} : \mathcal{X} \to \mathbb{R}^d$. We learn this embedding function $g_{\text{input}}$ jointly with an embedding function $g_{\text{text}} : \mathcal{T} \to \mathbb{R}^d$ that embeds text in the same space as the inputs.

2. **Slice** (Section 5.2): We identify underperforming regions in the cross-modal embedding space using an error-aware mixture model fit on the input embeddings $\mathbf{z}_{\text{input}} := \{z_i := g_{\text{input}}(x_i)\}_{i=1}^n$, model predictions $\{\hat{y}_i := h_\theta(x_i)\}_{i=1}^n$, and true class labels $\{y_i\}_{i=1}^n$. This yields $\hat{k}$ slicing functions of the form $\psi_{\text{slice}}^{(j)} : \mathcal{X} \times \mathcal{Y} \to \{0, 1\}$.

3. **Describe** (Section 5.3): Finally, we use the text embedding function $g_{\text{text}}$ learned in step (1) to generate a set of $\hat{k}$ natural language descriptions of the discovered slices.

### 5.1 EMBEDDING INPUTS WITH CROSS-MODAL REPRESENTATIONS

Cross-modal representation learning algorithms embed input examples and paired text data in the same latent representation space. Formally, given a dataset of paired inputs $V \in \mathcal{X}$ and text descriptions $T \in \mathcal{T}$, $\mathcal{D}_{\text{paired}} = \{(v_i, t_i)\}_{i=1}^{n_{paired}}$, we learn two embedding functions $g_{\text{input}} : \mathcal{X} \to \mathbb{R}^d$ and $g_{\text{text}} : \mathcal{T} \to \mathbb{R}^d$ such that the distances between pairs of embeddings $dist(g_{\text{input}}(v_i), g_{\text{text}}(t_j))$ reflect the semantic similarity between $v_i$ and $t_j$ for all $i, j \leq n_{paired}$.

Ultimately, this joint training procedure enables the creation of semantically meaningful input embeddings that incorporate information from text. In this work, our key insight is that input representations generated from cross-modal learning techniques encode the semantic knowledge necessary for identifying coherent slices. Our method relies on the assumption that we have access to either (a) pretrained cross-modal embedding functions or (b) a dataset with paired input-text data that can be used to learn cross-modal embedding functions. It is important to note that paired data is only required if a practitioner wishes to generate custom cross-modal embedding functions.

Domino uses four types of cross-modal embeddings to enable slice discovery across our input domains: CLIP (Radford et al., 2021), ConVIRT (Zhang et al., 2020), MIMIC-CLIP, and EEG-CLIP. We adapt CLIP and ConVIRT from prior work, and we train MIMIC-CLIP and EEG-CLIP on large datasets with paired inputs and text (implementation details are provided in Section A.4.1).

### 5.2 CLUSTERING EMBEDDINGS WITH ERROR-AWARE MIXTURE MODEL

We then proceed to the second step in the Domino pipeline: slicing. Recall that our goal is to find a set of $\hat{k}$ slicing functions that partition our data into coherent and underperforming slices. Taking inspiration from the recently developed *Spotlight* algorithm (d'Eon et al., 2021), we propose

a mixture model that jointly models the input embeddings, class labels, and model predictions. This encourages slices that are homogeneous with respect to error type (*e.g.* all false positives). The model assumes that data is generated according to the following generative process: each example is randomly assigned membership to a *single* slice according to a categorical distribution $\mathbf{S} \sim Cat(\mathbf{p_S})$ with parameter $\mathbf{p_S} \in \{\mathbf{p} \in \mathbb{R}_+^{\bar{k}} : \sum_{i=1}^{\bar{k}} p_i = 1\}$. Given membership in slice $j$, the embeddings are normally distributed $Z|S^{(j)} = 1 \sim \mathcal{N}(\mu^{(j)}, \boldsymbol{\Sigma}^{(j)})$ with parameters mean $\mu^{(j)} \in \mathbb{R}^d$ and covariance $\boldsymbol{\Sigma}^{(j)} \in \mathbb{S}_{++}^d$ (the set of symmetric positive definite $d \times d$ matrices), the labels vary as a categorical $Y|S^{(j)} = 1 \sim Cat(\mathbf{p}^{(j)})$ with parameter $\mathbf{p}^{(j)} \in \{\mathbf{p} \in \mathbb{R}_+^c : \sum_{i=1}^c p_i = 1\}$, and the model predictions also vary as a categorical $\hat{Y}|S^{(j)} = 1 \sim Cat(\hat{\mathbf{p}}^{(j)})$ with parameter $\hat{\mathbf{p}}^{(j)} \in \{\hat{\mathbf{p}} \in \mathbb{R}_+^c : \sum_{i=1}^c \hat{p}_i = 1\}$. This assumes that the embedding, label, and prediction are all independent conditioned on the slice.

The mixture model is parameterized by $\phi = [\mathbf{p}_S, \{\mu^{(j)}, \Sigma^{(j)}, \mathbf{p}^{(j)}, \hat{\mathbf{p}}^{(j)}\}_{j=1}^{\bar{k}}]$. The log-likelihood over the validation dataset $D_v$ is given as follows and maximized using expectation-maximization:

$$\ell(\phi) = \sum_{i=1}^n \log \sum_{j=1}^{\bar{k}} P(S^{(j)}=1)P(Z=z_i|S^{(j)}=1)P(Y=y_i|S^{(j)}=1)^\gamma P(\hat{Y}=h_\theta(x_i)|S^{(j)}=1)^\gamma,$$

(1)

where $\gamma$ is a hyperparameter that balances the trade-off between coherence and under-performance, our two desiderata (see Section A.4.2).

## 5.3 Generating Natural Language Descriptions of Discovered Slices

Domino can produce natural language statements that describe characteristics shared between examples in the discovered slices. To generate slice descriptions, we begin by sourcing a corpus of candidate natural language phrases $\mathcal{D}_{\text{text}} = \{t_j\}_{j=1}^{n_{\text{text}}}$. Critically, this text data can be sourced independently and does not need to be paired with examples in $\mathcal{D}$. In Section A.4.3, we describe an approach for generating a corpus of phrases relevant to the domain.

We generate an embedding for each phrase in $\mathcal{D}_{\text{text}}$ using the cross-modal embedding function $g_{\text{text}}$, yielding $\{z_j^{\text{text}} := g_{\text{text}}(t_j)\}_{j=1}^{n_{\text{text}}}$. Then, we compute a prototype embedding for each of the discovered slices by taking the weighted average of the input embeddings in the slice, $\{\bar{z}_{\text{slice}}^{(i)} := \psi_{\text{slice}}^{(i)}(\mathbf{x}, \mathbf{y})^\top \mathbf{z}_{\text{input}}\}_{i=1}^{\hat{k}}$. We also compute a prototype embedding for each class $\{\bar{z}_{\text{class}}^{(c)} := \frac{1}{n_c} \sum_{i=1}^n \mathbf{1}[y_i = c] z_i^{\text{input}}\}_{c=1}^C$. To distill the slice prototypes, we subtract out the prototype of the most common class in the slice $\bar{z}_{\text{slice}}^{(i)} - \bar{z}_{\text{class}}^{(c)}$. Finally, to find text that describes each slice, we compute the dot product between the distilled slice prototypes and the text embeddings and return the phrase with the highest value: $\text{argmax}_{j \in [n_{\text{text}}]} z_j^{\text{text}\top} (\bar{z}_{\text{slice}}^{(i)} - \bar{z}_{\text{class}}^{(c)})$.

## 6 Experiments

We use the evaluation framework developed in Section 4 to systematically assess Domino, comparing it to existing SDMs across 1,235 slice discovery settings. Our experiments validate the three core design choices behind Domino: (1) the use of cross-modal embeddings, (2) the use of a novel error-aware mixture model, and (3) the generation of natural language descriptions for slices. We provide SDM implementation details in Section A.3.2 and extended evaluations in Section A.5.

### 6.1 Cross-Modal Embeddings Improve SDM Performance

In this section, we evaluate the effect of embedding type on performance. We hold our error-aware slicing algorithm (Step 2) constant and vary our choice of embedding (Step 1).

**Natural Images.** We compare four embeddings: final-layer activations of a randomly-initialized ResNet-50 (He et al., 2016), final-layer activations of $h_\theta$, BiT (Kolesnikov et al., 2019), and CLIP (Radford et al., 2021). CLIP embeddings are cross-modal. Results are shown in Figure 3.

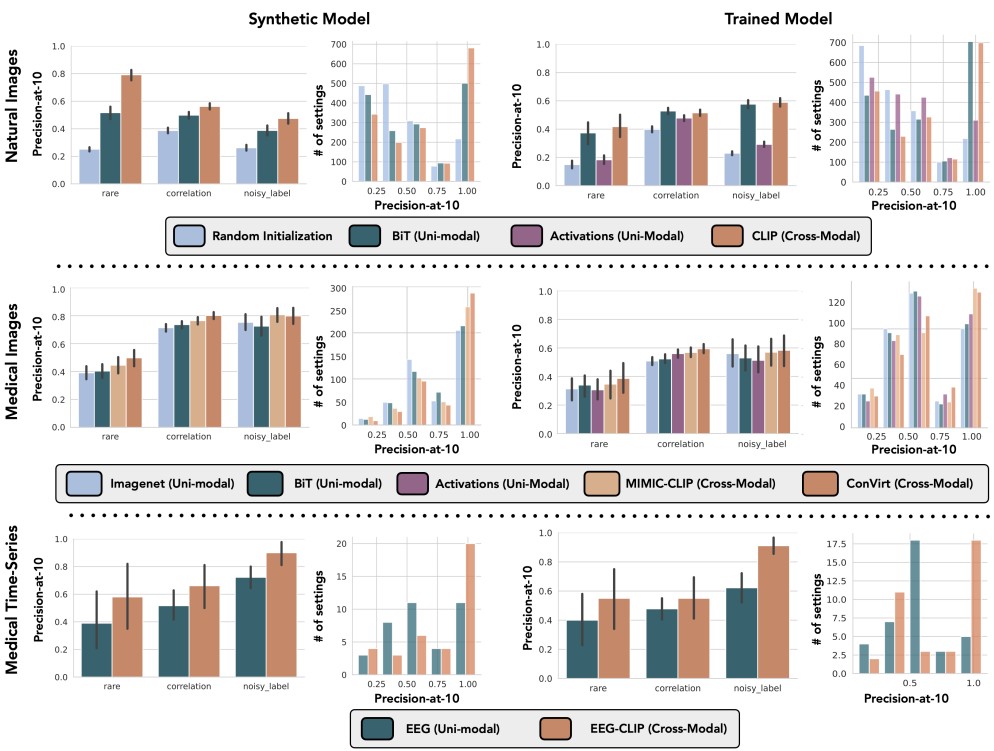

Figure 3: **Cross-modal embeddings enable accurate slice discovery**. Using our evaluation framework, we demonstrate that the use of cross-modal embeddings leads to consistent improvements in slice discovery across three datasets and two input modalities (1,235 settings).

When evaluating with synthetic models, we find that using CLIP embeddings results in a mean precision-at-10 of 0.570 (95% CI: 0.554, 0.586), a 9 percentage point increase over BiT embeddings and a 23 percentage point increase over random activations.[2]

When evaluating with trained models, we find no difference between using CLIP embeddings and BiT embeddings. However, both outperform activations of the trained classifier $h_\theta$ by nearly 15 percentage points in mean precision-at-10. This finding is of particular interest given that classifier activations are a popular embedding choice in prior SDMs (d'Eon et al., 2021; Sohoni et al., 2020). Notably, the gap between CLIP and $h_\theta$ activations is much smaller in settings with correlation slices. This makes sense because a model that relies on a correlate to make predictions will likely capture information about the correlate in its activations (Sohoni et al., 2020).

**Medical Images.** We compare five embeddings: the final-layer activations of a ResNet-50 pretrained on ImageNet (He et al., 2016), the final-layer activations of the trained classifier $h_\theta$, BiT (Kolesnikov et al., 2019), and domain-specific cross-modal embeddings that we trained using two different methods: MIMIC-CLIP and ConVIRT. For synthetic models, cross-modal ConVIRT embeddings enable a mean precision-at-10 of 0.765 (95% CI: 0.747, 0.784), a 7 point improvement over the best unimodal embeddings (BiT) with mean precision-at-10 of 0.695 (95% CI: 0.674, 0.716). For trained models, we again find that although $h_\theta$ activations perform the worst on rare and noisy label slices, they are competitive with cross-modal embeddings on correlation slices.

**Medical Time Series.** For our EEG dataset, we compare the final-layer activations of a pretrained seizure classifier and a CLIP-style cross-modal embedding trained on EEG-report pairs. When evaluating with synthetic models, we find that cross-modal embeddings recover coherent slices with a mean precision-at-10 of 0.697 (95% CI: 0.605, 0.784). This represents a 17 point gain over using unimodal embeddings 0.532 (95% CI: 0.459, 0.608). Cross-modal embeddings also outperform unimodal embeddings when evaluated with trained models. This demonstrates that cross-modal embeddings can aid in recovering coherent slices even in input modalities other than images.

---

[2]Note that synthetic models do not have activations, so we cannot compare to trained activations here.

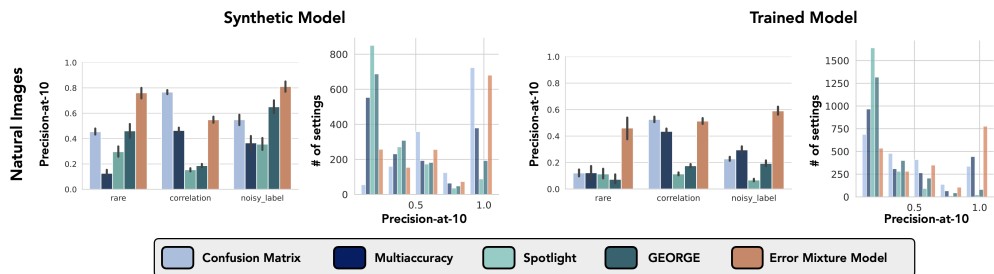

Figure 4: **Error-aware mixture model enables accurate slice discovery**. When cross-modal embeddings are provided as input, our error-aware mixture model often outperforms previously-designed SDMs. Results on medical images and medical time-series data are in Section A.5.

## 6.2 ERROR-AWARE MIXTURE MODEL IMPROVES SDM PERFORMANCE

In order to understand the effect of slicing algorithms on performance, we hold the cross-modal embeddings (Step 1) constant and vary the slicing algorithm (Step 2). We compare the error-aware mixture model to four prior SDMs: George (Sohoni et al., 2020), Multiaccuracy (Kim et al., 2018), Spotlight (d'Eon et al., 2021), and a baseline we call ConfusionSDM, which outputs slicing functions that partition data into the cells of the confusion matrix. We provide cross-modal embeddings as input to all five SDMs. On noisy and rare slices in natural images, the error-aware mixture model recovers ground truth slices with a mean precision-at-10 of 0.639 (95% CI: 0.617,0.660) – this represents a 105% improvement over the next-best method, George. Interestingly, on correlation slices, the naive ConfusionSDM baseline outperforms our error-aware mixture model. Extended evaluations are provided in Section A.5.

## 6.3 DOMINO GENERATES NATURAL LANGUAGE DESCRIPTIONS OF DISCOVERED SLICES.

Domino is the first SDM that can generate natural language descriptions for identified slices. For natural images, we provide a quantitative analysis of these descriptions. Specifically, since Domino returns a ranking over all phrases in the corpus $\mathcal{D}_{\text{text}}$, we can compute the percentage of settings in which the name of the "ground truth" slice (or a WordNet synonym (Fellbaum, 1998)) appears in the top-$k$ words returned by Domino. In Figure 9, we plot this percentage for $k = 1$ to $k = 10$. We find that for 34.7% of rare slices, 41.0% of correlation slices, and 39.0% of noisy label slices, Domino ranks the name of the slice (or a synonym) first out of the thousands of phrases in our corpus. In 57.4%, 55.4%, and 48.7% of rare, correlation, and noisy label slices respectively, Domino ranks the phrase in the top ten. In Section A.1, we show that Domino is successful at recovering accurate explanations for natural image, medical image, and medical time series data.

## 7 CONCLUSION

In this work, we analyze the slice discovery problem. First, we observe that existing approaches for evaluating SDM performance do not allow for large-scale, quantitative evaluations. We address this challenge by introducing a programmable framework to measure SDM performance across two axes: underperformance and coherence. Second, we propose Domino, a novel SDM that combines cross-modal representations with an error-aware mixture model. Using our evaluation framework, we demonstrate that the embedding and slicing steps of Domino outperform those of existing SDMs. We also show for the first time that using cross-modal embeddings for slice discovery can enable the generation of semantically meaningful slice descriptions. Notably, Domino requires only black-box access to models and can thus be broadly useful in settings where users have API access to models. Future directions include executing controlled user studies to evaluate when generated explanations are actionable, developing strategies for computing input embeddings when access to both pre-trained cross-modal embeddings and paired input-text data is limited, and exploring strategies for improving slice discovery in settings where slice strength, $\alpha$, is low. We hope that our evaluation framework accelerates the development of slice discovery methods and that Domino will help practitioners better evaluate their models.

## 8 REPRODUCIBILITY STATEMENT

We provide an open-source implementation of our evaluation framework at `https://github.com/HazyResearch/domino`. Users can run Domino on their own models and datasets by installing our Python package via: `pip install domino`.

## 9 ETHICS STATEMENT

Domino is a tool for identifying systematic model errors. No matter how effective it is at this task, there may still be failure-modes Domino will not catch. There is a legitimate concern that model debugging tools like Domino could give practitioners a false sense of security, when in fact their models are failing on important slices not recovered by Domino. It is critical that practitioners still run standard evaluations on accurately-labeled, representative test sets in addition to using Domino for auditing models. Additionally, because Domino uses embeddings trained on image-text pairs sourced from the web, it may reflect societal biases when identifying and describing slices. Future work should explore the impacts of using biased embeddings to identify errors in models. What kinds of error modes might we miss? Are certain underrepresented groups or concepts less likely to be identified as an underperforming slice?

## 10 CONTRIBUTIONS

S.E., M.V., K.S., J.D., J.Z., and C.R. conceptualized the overall study. S.E., M.V., K.S., J.-B.D., J.D., J.Z., and C.R. contributed to the experimental design while S.E., M.V., K.S., and J.-B.D. wrote computer code and performed experiments. S.E. trained all models for CelebA and ImageNet, M.V. trained all models for MIMIC, K.S. trained all models for EEG, and J.-B.D generated ConVIRT embeddings for MIMIC. C.L.-M. provided the labeled EEG data and clinical expertise. S.E., M.V., and K.S. prepared the manuscript. All authors contributed to manuscript review.

## 11 ACKNOWLEDGEMENTS

We are thankful to Karan Goel, Laurel Orr, Michael Zhang, Sarah Hooper, Neel Guha, Megan Leszczynski, Arjun Desai, Priya Mishra, Simran Arora, Jure Leskovec, Zach Izzo, Nimit Sohoni, and Weixin Liang for helpful discussions and feedback. Sabri Eyuboglu is supported by the National Science Foundation Graduate Research Fellowship. Maya Varma is supported by graduate fellowship awards from the Department of Defense (NDSEG) and the Knight-Hennessy Scholars program at Stanford University. Khaled Saab is supported by the Stanford Interdisciplinary Graduate Fellowship with the Wu Tsai Neurosciences Institute. James Zou is supported by NSF CAREER 1942926. We gratefully acknowledge the support of NIH under No. U54EB020405 (Mobilize), NSF under Nos. CCF1763315 (Beyond Sparsity), CCF1563078 (Volume to Velocity), and 1937301 (RTML); ARL under No. W911NF-21-2-0251 (Interactive Human-AI Teaming); ONR under No. N000141712266 (Unifying Weak Supervision); ONR N00014-20-1-2480: Understanding and Applying Non-Euclidean Geometry in Machine Learning; N000142012275 (NEPTUNE); NXP, Xilinx, LETI-CEA, Intel, IBM, Microsoft, NEC, Toshiba, TSMC, ARM, Hitachi, BASF, Accenture, Ericsson, Qualcomm, Analog Devices, Google Cloud, Salesforce, Total, the HAI-GCP Cloud Credits for Research program, the Stanford Data Science Initiative (SDSI), and members of the Stanford DAWN project: Facebook, Google, and VMWare. The U.S. Government is authorized to reproduce and distribute reprints for Governmental purposes notwithstanding any copyright notation thereon. Any opinions, findings, and conclusions or recommendations expressed in this material are those of the authors and do not necessarily reflect the views, policies, or endorsements, either expressed or implied, of NIH, ONR, or the U.S. Government.

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

# A  APPENDIX

## CONTENTS

## A.1 EXAMPLES OF SLICE DESCRIPTIONS

In this section, we provide examples of the natural language slice descriptions generated by Domino. Figure 5 includes natural language descriptions and representative photos for discovered slices in the natural image domain, Figure 6 includes natural language descriptions in the medical image domain, and Figure 7 includes natural language descriptions in the medical time-series domain.

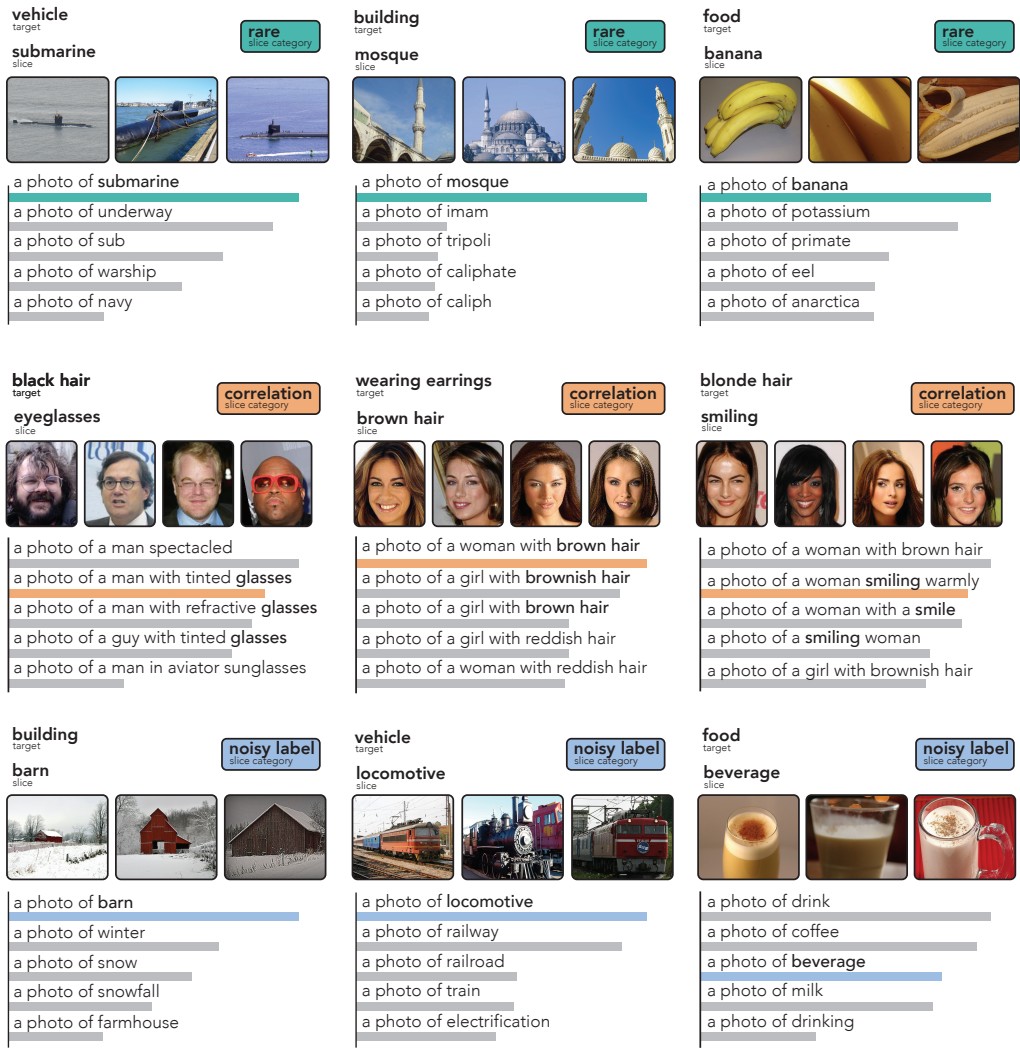

Figure 5: **Domino produces natural language descriptions of discovered slices**. Natural language descriptions for discovered slices in (top row) 3 settings randomly selected from the set of the 85 rare slice, natural image settings where Domino includes the exact name of the slice in its top 5 slice descriptions; (middle row) 3 settings randomly selected from the set of the 45 correlation slice, natural image settings where Domino includes the exact name of the slice in its top 5 slice descriptions and precision-at-25 exceeds 0.8; and (bottom row) 3 settings randomly selected from the set of the 95 noisy label slice, natural image settings where Domino includes the exact name of the slice in its top 5 slice descriptions and precision-at-25 exceeds 0.8. The length of the bars beneath each description are proportional to the dot product score for the description (see Section 5.3). Also shown are the top 3-4 images that Domino associates with the discovered slice.

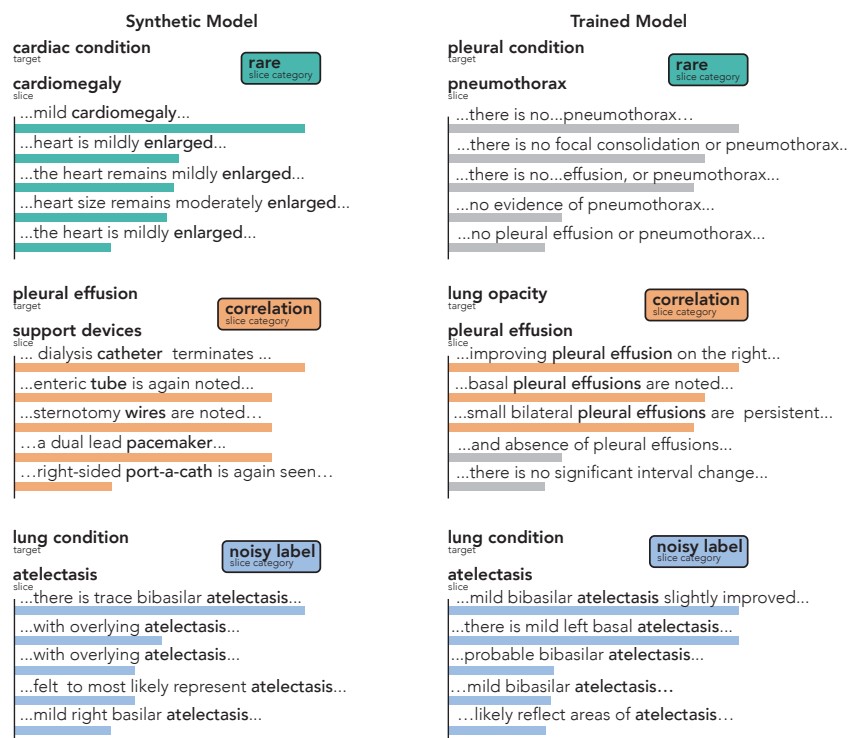

Figure 6: **Domino produces natural language descriptions for discovered slices in a medical image dataset**. Here, we provide the top five natural language descriptions for discovered slices in (top row) two rare slice settings, (middle row) two correlation slice settings, and (bottom row) two noisy label slice settings. Colored bars represent accurate slice descriptions, and gray bars represent incorrect descriptions. Note that in our medical examples, the vocabulary consists of physician reports in the training set; since, we are unable to provide the full-text reports due to patient privacy concerns, the figure includes relevant fragments of reports. The length of the bars beneath each description are proportional to the dot product score for the description (see Section 5.3)

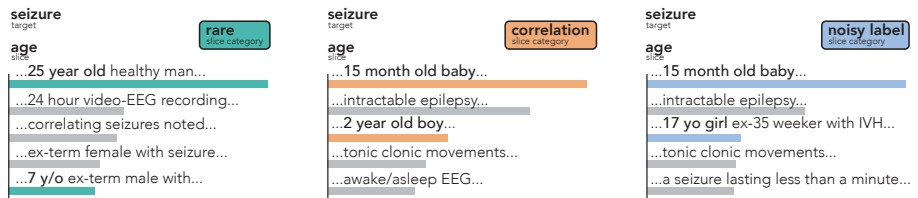

Figure 7: **Domino produces natural language explanations for discovered slices in a medical time-series dataset**. Natural language descriptions for discovered slices in 3 of our EEG slice discovery settings. In all three settings, the model is trained to detect seizures and underperforms on the slice of young patients. The three settings span all three slice categories and in each Domino describes the slice with reports mentioning young age. Note that the description corpus consists of physician reports in the training set; however, we are unable to provide the full-text reports due to patient privacy concerns, so the table includes only relevant fragments. The bar lengths beneath each description are proportional to the dot product score for the description (see Section 5.3)

## A.2 Extended Related Work: Survey of Slices in the Wild

Several recent studies have shown that machine learning models often make systematic errors on critical data slices. In this section, we provide a survey of underperforming slices documented in the literature.

- **Skin Lesion Classification (Correlation Slice):** Bissoto et al. (2019) reveal that models trained to classify skin lesion images depend on clinically irrelevant information due to biases in the training data. Specifically, a computer vision model trained to classify skin lesions performs poorly on images of malignant lesions with color charts (*i.e.* colored bandages), since color charts more commonly appear with benign lesions.

- **Melanoma Detection (Correlation Slice):** A study by Winkler et al. (2019) showed that melanoma detection models trained on dermascopic image datasets often rely on the presence of surgical skin markings when making predictions. Since dermatologists mark suspicious lesions during clinical practice with a gentian violet surgical skin marker, dermascopic images will often include skin markings, causing models to learn spurious correlations between markings and the presence of melanoma. Models then underperform on the slice of lesions without markings.

- **Pneumothorax Detection (Correlation Slice):** Models trained to detect the presence of pneumothorax (collapsed lungs) have been found to rely on the presence of chest drains, a device used during treatment (Oakden-Rayner et al., 2019).

- **Hip-Fracture Detection (Rare Slice):** Due to the low prevalence of cervical fractures in a pelvic X-ray dataset collected from the Royal Adelaide Hospital, computer vision models trained to detect fractures underperform on this slice (Oakden-Rayner et al., 2019).

- **Hip-Fracture Detection (Correlation Slice):** Badgeley et al. (2019) show that the performance of models trained to detect hip fractures from X-rays are sensitive to multiple patient-specific and hospital-specific attributes. In particular, when the test distribution is subsampled to remove correlations with patient attributes (age, gender, BMI, pain, fall) and hospital attributes (scanner, department, radiation, radiologist name, order time), the model performance drops to close to random.

- **Gender Classification in Images (Rare Slice):** Buolamwini & Gebru (2018) demonstrated that facial analysis datasets are often composed primarily of lighter-skinned subjects (based on Fitzpatrick Skin Types) and as a result, three commercial gender classification systems systematically underperform on rare subgroups (*e.g.* darker faces, female faces).

- **COVID-19 Detection in Chest X-rays (Correlation Slice):** DeGrave et al. (2021) reveal that some models trained to detect COVID-19 from radiographs do not generalize to datasets from external hospitals, indicating that the models rely on source-specific attributes instead of pathology markers.

- **Pneumonia Detection in Chest X-rays (Correlation Slice):** Zech et al. (2018) evaluated pneumonia screening CNNs on three external hospitals, and found that performance on the external hospitals was significantly lower than the original hospital dataset. Additionally, the CNNs were able to very accurately classify the hospital system and department where the radiographs were acquired, indicating that the CNN features had learned hospital-specific confounding variables.

- **Weakly Supervised Aortic Valve Malformation Classification (Noisy Label Slice):** Weak supervision is commonly used in medical machine learning practice to label clinical datasets. Using a set of labeling functions, Fries et al. (2019) train a weakly-supervised model to classify aortic valve malformations. They note that noise in the labels is induced by labeling functions. These may systematically miss coherent slices of data, meaning that a model trained on these labels may underperform on those slices.

- **Predicting Gender from Photos of the Iris (Correlation Slice):** Several studies have reported models capable of predicting a person's gender from a photo of their iris Bansal et al. (2012); Tapia et al. (2016). However, Kuehlkamp et al. (2017) show that these models may be relying on the presence of mascara to make these predictions. This suggests that performance will likely be degraded on the slice of females without mascara and males with mascara.

- **Speech Recognition (Rare Slice):** Koenecke et al. (2020) demonstrated that automated speech recognition systems have large performance disparities between white and African American speakers. The disparities were traced to the race gap in the corpus used to train the model, indicating that African American speakers were a rare slice.

- **Object Recognition (Rare Slice):** A study by de Vries et al. (2019) demonstrated that publicly available object recognition algorithms often systematically underperform on household items that commonly occur in non-Western countries and low-income communities. This is likely due to objects appearing in different environments as well as differences in object appearance.

- **Named Entity Disambiguation (Rare Slice)**: Named entity disambiguation (NED) systems, which map textual mentions to structured entities, play a critical role in automated text parsing pipelines. Several studies have demonstrated that NED systems underperform on rare entities that occur infrequently in the training data (Orr et al., 2020; Varma et al., 2021).

## A.3   EXTENDED DESCRIPTION OF EVALUATION FRAMEWORK

| Domain | Rare Settings | Correlation Settings | Noisy Label Settings | Total settings |
|--------|---------------|----------------------|----------------------|----------------|
| Natural Images | 177 (trained) 197 (synthetic) | 520 (trained) 520 (synthetic) | 287 (trained) 394 (synthetic) | 984 (trained) $1,111$ (synthetic) |
| MIMIC | 15 (trained) 55 (synthetic) | 176 (trained) 352 (synthetic) | 30 (trained) 55 (synthetic) | 221 (trained) 462 (synthetic) |
| EEG | 10 (trained) 10 (synthetic) | 10 (trained) 10 (synthetic) | 10 (trained) 10 (synthetic) | 30 (trained) 30 (synthetic) |

Table 1: **Overview of evaluation framework.** We evaluate SDMs across 1,235 (trained) slice discovery settings across three domains, four slice categories, and five slice parameters ($\alpha$).

### A.3.1   EXTENDED DESCRIPTION OF SLICE CATEGORIES

In Section 4.1.1, we describe how we categorize each slice discovery setting based on the underlying reason that the model $h_\theta$ exhibits degraded performance on the slices **S**. We survey the literature for examples of underperforming slices in the wild, which we document in Section A.2. Based on our survey and prior work (Oakden-Rayner et al., 2019), we identify three popular slice types. We provide expanded descriptions below.

**Rare slice.** Consider a slice $S \in \{0, 1\}$ (*e.g.* patients with a rare disease, photos taken at night) that occurs infrequently in the training set (*i.e.* $P(S = 1) < \alpha$ for some small $\alpha$). Since a rare slice will not significantly affect model loss during training, the model may fail to learn to classify examples within the slice. To generate settings with rare slices, we use base datasets with hierarchical label schema (*e.g.* ImageNet (Deng et al., 2009)). We construct dataset $\mathcal{D}$ such that for a given class label $Y$, the elements in subclass $C$ occur with proportion $\alpha$, where $\alpha$ ranges between 0.01 and 0.1.

**Correlation slice.** If the target variable $Y$ (*e.g.* pneumothorax) is correlated with another variable $C$ (*e.g.* chest tubes), the model may learn to rely on $C$ to make predictions. This will induce a slice $S = \mathbf{1}[C \neq Y]$ (*e.g.* pneumothorax without chest tubes and normal with chest tubes) with degraded performance. To generate settings with correlation slices, we use base datasets with metadata annotations (*e.g.* CelebA (Liu et al., 2015)). We sub-sample the base dataset $\mathcal{D}_{\text{base}}$ such that the resulting dataset $\mathcal{D}$ exhibits a linear correlation of strength $\alpha$ between the target variable and another metadata label $C$. Here, $\alpha$ ranges between 0.2 and 0.8.

We now describe our procedure for subsampling the base dataset to achieve a desired correlation. Assume we have two binary variables $Y, C \in \{0, 1\}$ (Bernoulli random variables) and a dataset of $\mathcal{D}_{\text{base}} = \{(y_i, c_i)\}_{i=1}^{n_{\text{base}}}$. Given a target correlation $\alpha$, we would like to subsample the dataset $\mathcal{D}_{\text{base}}$ such that the resulting dataset $D = (y_i, c_i)_{i=1}^{n}$ of size $n$ exhibits a sample correlation between $Y$ and $C$ of $\alpha$.

The population correlation between $Y$ and $C$ is given by $\alpha(Y,C) = \frac{\text{cov}(Y,C)}{\sigma_Y \sigma_C}$ and $\text{cov}(Y,C) = \mathbb{E}[YC] - E[Y]E[C]$. For a sample, the unbiased estimator of the covariance is:

$$\text{cov}(\mathbf{y},\mathbf{c}) = \frac{1}{n-1}\sum_{i=1}^{n}(y_i - \bar{y})(c_i - \bar{c}) = \frac{1}{n-1}\left(\sum_{i=1}^{n} y_i c_i - \bar{c}\bar{y}\right)$$

Since we know $Y$ and $C$ are Bernoulli random variables, we can express this in terms of variables like $n_{y=1,c=1}$ (*i.e.* the number of samples $i$ where $y_i = 1$ and $c_i = 1$) and $n_{y=1}$(*i.e.* the number of samples $i$ where $y_i$=1).

$$\text{cov}(\mathbf{y},\mathbf{c}) = \frac{1}{n-1}\left(n_{y=1,c=1} - \frac{n_{y=1}n_{c=1}}{n}\right)$$

The correlation coefficient can then be expressed in terms of this covariance and the sample standard deviations $s_y$ and $s_c$:

$$r = \frac{n_{y=1,c=1} - \frac{n_{y=1}n_{c=1}}{n}}{(n-1)s_y s_c}$$

In addition to supplying a target correlation $\alpha$, assume we also have target means $\mu_a$ and $\mu_b$ (these could be the sample means in the original dataset $\mathcal{D}$ for example) and a target sample size $n$. Since Y and C are Bernoulli random variables, we can compute sample standard deviations as $s_y = \mu_a(1 - \mu_a)$ and $s_c = \mu_b(1 - \mu_b)$. We can then derive simple formulas for computing the desired values needed to properly subsample the data:

$$n_{y=1} = \mu_a n$$
$$n_{c=1} = \mu_b n$$
$$n_{y=1,c=1} = \alpha(n-1)s_y s_c + \frac{n_{y=1}n_{c=1}}{n}$$
$$n_{y=1,c=0} = n_{y=1} - n_{y=1,c=1}$$
$$n_{y=0,c=1} = n_{c=1} - n_{y=1,c=1}$$
$$n_{y=0,c=0} = n - (n_{y=1} + n_{c=1} - n_{y=1,c=1})$$

**Noisy label slice.** Errors in labels are not always distributed uniformly across the training distribution. A slice of data $S \in \{0,1\}$ may exhibit higher label error rates than the rest of the training distribution. This could be due to a number of different factors: classes may be ambiguous (*e.g.* annotators labeling sandwiches may disagree whether to include hot dogs), labeling heuristics may fail on important slices (*e.g.* medical imaging heuristics that only activate on images from one scanner type), or human annotators may lack expertise on certain subsets (*e.g.* annotators from one country labeling street signs in another). A model $h_\theta$ trained on these labels will likely exhibit degraded performance on $S$. To generate settings with noisy label slices, we can use a base dataset with metadata annotations (*e.g.* CelebA (Liu et al., 2015)). We construct dataset $D$ such that for each class label $Y$, the elements in subclass $C$ exhibit label noise with probability $\alpha$, where $\alpha$ ranges between 0.01 and 0.3.

### A.3.2 SDM IMPLEMENTATIONS

**Spotlight.** d'Eon et al. (2021) search for an underperforming slice by maximizing the expected loss with respect to a multivariate Gaussian with spherical covariance. We use the authors' implementation provided at `https://github.com/gregdeon/spotlight`. We enforce a minimum spotlight size equal to 2% of the validation data as recommended in the paper. We use a initial learning rate of $1 \times 10^{-3}$ (the default in the implementation) and apply the same annealing and barriers as the authors. We optimize for 1,000 steps per spotlight (the default in the implementation).

**GEORGE.** Sohoni et al. (2020) propose identifying underperforming slices by performing class conditional clustering on a U-MAP reduction of the embeddings. We use the implementation provided at `https://github.com/HazyResearch/hidden-stratification`.

**Multiaccuracy Boost.** To identify slices where the model $h_\theta$ is systematically making mistakes, Kim et al. (2018) use ridge regression to learn a function $f : \mathbb{R}^d \to \mathbb{R}_+$ mapping from an example's embedding $z_i \in \mathbb{R}^d$ to the partial derivative of the cross entropy loss with respect to the prediction

$$\frac{\partial \ell(h_\theta(x_i), y_i)}{\partial h_\theta(x_i)} = \frac{1}{1 - h_\theta(x_i) - y_i}. \tag{2}$$

Because this function grows as the absolute value of the residual $|h_\theta(x_i) - y_i|$ grows, a good $f$ should correlate with the residual.

In order to discover multiple distinct slices, the authors repeat this process $\hat{k}$ times updating the model predictions on each iteration according to

$$h_\theta^{(j+1)}(x_i) \propto e^{-\eta f(z_i)} h_\theta^{(j)}(x_i), \tag{3}$$

where $\eta$ is a hyperparameter defining the step size for the update.

We use an implementation of Multiaccuracy Boost based on the authors', which was released at `https://github.com/amiratag/MultiAccuracyBoost`. We use $\eta = 0.1$ as in the authors' implementation. We fit $f$ on 70% of the validation data and use the remaining 30% for evaluating the correlation with the residual. The authors use the same ratio in their implementation.

**Confusion SDM.** A simple, embedding-agnostic way to identify underperforming subgroups is to simply inspect the cells of the confusion matrix. We include this important baseline to determine when more complicated slice discovery techniques are actually useful.

### A.3.3 Extended Description of Trained Models

In Section 4.1.2, we discuss training a distinct model $h_\theta$ for each slice discovery setting. In this section we provide additional details on model training.

For our **natural image** settings and **medical image** settings, we used a ResNet-18 randomly initialized with He initialization (He et al., 2015; 2016). We applied an Adam optimizer with learning rate $1 \times 10^{-4}$ for 10 epochs and use early stopping using the validation dataset (Kingma & Ba, 2017). During training, we randomly crop each image, resize to $224 \times 224$, apply a random horizontal flip, and normalize using ImageNet mean and standard deviation ($\mu = [0.485, 0.456, 0.406]$, $\sigma = [0.229, 0.224, 0.225]$). During inference, we resize to $256 \times 256$, apply a center crop of size $224 \times 224$ and normalize with the same mean and standard deviation as in training.

For our **medical time series** settings, we use a densely connected inception convolution neural network (Roy et al., 2019) randomly initialized with He initialization (He et al., 2015; 2016). Since the EEG signals are sampled at 200 Hz, and the EEG clip length is 12 seconds, with 19 EEG electrodes, the input EEG has shape $19 \times 2400$. The models are trained with a learning rate of $10^{-6}$ and a batch size of 16 for 15 epochs.

### A.3.4 Extended Description of Synthetic Models

In Section 4.1.2, we discuss how synthesizing model predictions in order to provide greater control over the evaluation. Here, we provide additional details describing this process.

Assume a binary class label $Y \in \{0, 1\}$ and a slice variable $S \in \{0, 1\}$. We sample the predicted probability $\hat{Y} \in [0, 1]$ from one of four beta distributions, conditional on $Y$ and $S$. The parameters of those four beta distributions are set so as to satisfy a desired specificity and sensitivity in the slice (*i.e.* when $S = 1$) and out of the slice (*i.e.* when $S = 0$).

For natural image settings, we set both specificity and sensitivity to $0.4$ in the slice and $0.75$ out of the slice. For medical image and time series settings, we set both specificity and sensitivity to $0.4$ in the slice and $0.8$ out of the slice.

## A.4 Extended Description of Domino

### A.4.1 Extended Description of Cross-Modal Embeddings

Here, we provide implementation details for the four cross-modal embeddings used in this work: CLIP, ConVIRT, MIMIC-CLIP, and EEG-CLIP. Domino relies on the assumption that we have ac-

cess to either (a) pretrained cross-modal embedding functions or (b) a dataset with paired input-text data that can be used to learn embedding functions.

Large-scale pretrained cross-modal embedding functions can be used to generate accurate representations of input examples. For instance, if our inference dataset consists of natural images, we can use a pre-trained CLIP model as embedding functions $g_{\text{input}}$ and $g_{\text{text}}$ to obtain image embeddings that lie in the same latent representation space as word embeddings.

However, pre-trained cross-modal embeddings are only useful if the generated representations accurately represent the inputs in the inference dataset. For example, if the inference dataset consists of images from specialized domains (*i.e.* x-rays) or non-image inputs, CLIP is likely to generate poor representations.

If pre-trained cross-modal embeddings are not available or cannot effectively represent the inference dataset, we require access to a separate dataset that can be used to learn the cross-modal embedding functions $g_{\text{input}}$ and $g_{\text{text}}$; we will refer to this dataset as the *CM-training dataset* for the remainder of this work. The CM-training dataset must consist of paired input-text data (*e.g.* image-caption pairs or radiograph-report pairs). Further, we assume that the text data provides a sufficient description of the input and includes information about potential correlates, such as object attributes or subject demographics. Note that paired input-text data is only required for the CM-training dataset; we make no such assumptions about the slice discovery dataset.

We use the following four cross-modal embeddings in our analysis:

- **CLIP (Natural Images)**: CLIP embeddings are cross-modal representations generated from a large neural network trained on 400 million image-text pairs (Radford et al., 2021).

- **ConVIRT (Medical Images)**: ConVIRT embeddings are generated from pairs of chest X-rays and radiologist reports in the MIMIC-CXR dataset. We create a CM-training set with 70% of the subjects in MIMIC-CXR, ensuring that no examples in our training set occur in validation or test data at slice discovery time. We then replicate the training procedure detailed in Zhang et al. (2020), which uses contrastive learning to align embeddings. We use the implementation provided in ViLMedic (Delbrouck et al., 2022).

- **MIMIC (Medical Images)**: We generate a separate set of cross-modal embeddings for the MIMIC-CXR dataset using a variant of the CLIP training procedure (Radford et al., 2021). We use the same CM-training set as detailed above, with $89,651$ image and report pairs. In order to generate text representations, we extract the findings and impressions sections from radiologist reports, which then serve as input to a BERT-based transformer initialized with weights from CheXBert and then frozen (Smit et al., 2020). Chest x-rays are passed as input to a visual transformer (ViT) pre-trained on ImageNet-21k and ImageNet 2012 at an image resolution of $224 \times 224$ (Dosovitskiy et al., 2021). All images are resized to $224 \times 224$ and normalized using mean and standard deviation values from ImageNet. Text representations are extracted from the output of the [CLS] token and image representations are extracted from the output of the final model layer. Image representation and text representations are separately passed through projection layers consisting of fully-connected layers and non-linear activation functions as detailed in an open-source implementation of CLIP[3]. Finally, we align text and image representations using the InfoNCE loss function with in-batch negatives as detailed in Radford et al. (2021). We train our implementation for 30 epochs with a learning rate of $10^{-4}$, a batch size of $64$, and an embedding dimension of $256$. The training process comes to an early stop if the loss fails to decrease for ten epochs.

- **EEG (Medical Time-Series Data)**: In order to generate cross-modal embeddings for EEG readings and associated neurologist reports, we modify the CLIP training procedure to work with time-series data (Radford et al., 2021; Saab et al., 2020). We create a CM-training set with $6,254$ EEG signals and associated neurologist reports. We use the same EEG encoder described in Section A.3.3. In order to represent text representations, we extract the findings and narrative from the reports, which are then served as input to a BERT-based transformer initialized with weights from CheXBert (Smit et al., 2020). Text and EEG representations are aligned using the InfoNCE loss function with in-batch negatives as described in (Radford et al., 2021). We also add a binary cross-entropy classification loss

---

[3]https://github.com/moein-shariatnia/OpenAI-CLIP

for seizure classification, where we weigh the InfoNCE loss by $0.9$, and the cross-entropy loss by $0.1$. The cross-modal model is trained with a learning rate of $10^{-6}$, an embedding dimension of 128, and a batch size of 32 for 200 epochs.

### A.4.2 Extended Description of Error-Aware Mixture Model

In Section 5.2, we describe a mixture model that jointly models the input embeddings, class labels, and model predictions. Here, we provide an expanded description of the model and additional implementation details.

**Motivation and relation to prior work.** Recall from Section 3 that our goal is to find a set of $\hat{k}$ slicing functions that partition our data into subgroups. This task resembles a standard unsupervised clustering problem but differs in an important way: we are specifically interested in finding clusters *where the model makes systematic prediction errors*. It is not immediately obvious how this constraint can be incorporated into out-of-the-box unsupervised clustering methods, such as principal component analysis or Gaussian mixture models. One potential approach would be to calculate the model loss on each example $x_i$ in $D_v$, append each loss value to the corresponding embedding $g_{\text{input}}(x_i)$, and cluster with standard methods (Sohoni et al., 2020). Empirically, we find that this approach often fails to identify underperforming slices, likely because the loss element is drowned out by the other dimensions of the embedding.

Recently, d'Eon et al. (2021) proposed *Spotlight*, an algorithm that searches the embedding space for contiguous regions with high-loss. Our error-aware mixture model is inspired by *Spotlight*, but differs in several important ways:

- Our model partitions the entire space, finding both high and low performing slices. In contrast, *Spotlight* searches only for regions with high loss. *Spotlight* introduces the "spotlight size" hyperparameter, which lower bounds the number of examples in the slice and prevents *Spotlight* from identifying very small regions.

- Because we model both the class labels and predictions directly, our error-aware mixture model tends to discover slices that are homogeneous with respect to error type. On the other hand, *Spotlight*'s objective is based on the loss, a function of the labels and predictions that makes false positives and false negatives indistinguishable (assuming cross-entropy).

- d'Eon et al. (2021) recommend using a spherical covariance matrix, $\mathbf{\Sigma} = aI$ with $a \in R$, when using *Spotlight* because it made their Adam optimization much faster and seemed to produce good results. In contrast, we use a diagonal covariance matrix of the form $\mathbf{\Sigma} = \text{diag}(a_1, a_2, ... a_{\hat{k}})$. Fitting our mixture model with expectation maximization remains tractable even with these more flexible parameters.

**Additional implementation details.** The mixture model's objective encourages both slices with a high concentration of mistakes as well as slices with a high concentration of correct predictions. However, in the slice discovery problem described in Section 3, the goal is only to identify slices of mistakes (*i.e.* slices that exhibit degraded performance with respect to some model $h_\theta$). To reconcile the model's objective with the goal of slice discovery, we model $\bar{k} > \hat{k}$ slices and then select $\hat{k}$ slices with the highest concentrations of mistakes. Specifically, we model $\bar{k} = 25$ slices and return the top $\hat{k}$ clusters with the largest absolute difference between $\hat{\mathbf{p}}$ and $\mathbf{p}$, $\sum_{i=1}^{c} |\hat{p}_i - p_i|$).

In practice, when $d$ is large (*e.g.* $d > 256$), we first reduce the dimensionality of the embedding to $d = 128$ using principal component analysis, which speeds up the optimization procedure significantly.

We include an important hyperparameter $\gamma \in \mathbb{R}_+$ that balances the importance of modeling the class labels and predictions against the importance of modeling the embedding. The log-likelihood over $n$ examples is given as follows and maximized using expectation-maximization:

$$\ell(\phi) = \sum_{i=1}^{n} \log \sum_{j=1}^{\bar{k}} P(S^{(j)}{=}1)P(Z{=}z_i|S^{(j)}{=}1)P(Y{=}y_i|S^{(j)}{=}1)^\gamma P(\hat{Y}{=}h_\theta(x_i)|S^{(j)}{=}1)^\gamma$$

(4)

In practice, this hyperparameter allows users to balance between the two desiderata highlighted in Section 1. When $\gamma$ is large (*i.e.* $\gamma > 1$), the mixture model is more likely to discover underper-

forming slices, potentially at the expense of coherence. On the other hand, when $\gamma$ is small (*i.e.* $0 \leq \gamma < 1$), the mixture model is more likely to discover coherent slices, though they may not be underperforming. In our experiments, we set $\gamma = 10$. We encourage users to tweak this parameter as they explore model errors with Domino, decreasing it when the discovered slices seem incoherent and increasing it when the discovered slices are not underperforming.

We initialize our mixture model using a scheme based on the confusion matrix. Typically, the parameters of a Gaussian mixture model are initialized with the centroids of a k-means fit. However, in our error-aware mixture model, we model not only embeddings (as Gaussians), but also labels and predictions (as categoricals). It is unclear how best to combine these variables into a single k-means fit. We tried initializing by just applying k-means to the embeddings, but found that this led to slices that were too heterogeneous with respect to error type (*e.g.* a mix of false positives and true positives). Instead, we use initial clusters where almost all of the examples come from the same cell in the confusion matrix. Formally, at initialization, each slice $j$ is assigned a $y^{(j)} \in \mathcal{Y}$ and $\hat{y}^{(j)} \in \mathcal{Y}$ (*i.e.* each slice is assigned a cell in the confusion matrix). This is typically done in a round-robin fashion so that there are at least $\lfloor \bar{k}/|\mathcal{Y}|^2 \rfloor$ slices assigned to each cell in the confusion matrix. Then, we fill in the initial responsibility matrix $Q \in \mathbb{R}^{n \times \bar{k}}$, where each cell $Q_{ij}$ corresponds to our model's initial estimate of $P(S^{(j)} = 1 | Y = y_i, \hat{Y} = \hat{y}_i)$. We do this according to

$$\bar{Q}_{ij} \leftarrow \begin{cases} 1 + \epsilon & y_i = y^{(j)} \wedge \hat{y}_i = \hat{y}^{(j)} \\ \epsilon & \text{otherwise} \end{cases} \qquad (5)$$

$$Q_{ij} \leftarrow \frac{\bar{Q}_{ij}}{\sum_{l=1}^{\bar{k}} \bar{Q}_{il}} \qquad (6)$$

where $\epsilon$ is random noise which ensures that slices assigned to the same confusion matrix cell won't have the exact same initialization. We sample $\epsilon$ uniformly from the range $[0, E]$ where $E$ is a hyperparameter set to $0.001$.

### A.4.3 GENERATING A CORPUS OF NATURAL LANGUAGE DESCRIPTIONS

In Section 5.3, we describe an approach for generating natural language descriptions of discovered slices. This approach uses cross-modal embeddings to retrieve descriptive phrases from a large corpus of text $\mathcal{D}_{\text{text}}$. In this section, we describe how we curate domain-specific corpora of descriptions to be used with Domino.

First, we solicit a set of *phrase templates* from a domain expert. For example, since CelebA is a dataset of celebrity portraits, we use templates like:

```
a photo of a person [MASK] [MASK]
a photo of a [MASK] woman
...
a [MASK] photo of a man
```

In our experiments with CelebA, we use a total of thirty templates similar to these (see GitHub for the full list). In our experiments with ImageNet we use only one template: `a photo of [MASK]`.

Next, we generate a large number of candidate phrases by filling in the `[MASK]` tokens using either (1) a pretrained masked-language model (in our experiments with CelebA we use a BERT base model (Devlin et al., 2018) and generate 100,000 phrases, keeping the $n_{\text{text}} = 10,000$) with the lowest loss) or (2) a programmatic approach (in our experiments with ImageNet, we create one phrase from each of the $n_{\text{text}} = 10,000$ most frequently used words on English Wikipedia (Semenov & Arefin, 2019)).

For our medical image and time-series datasets, we use corpora of physician reports sourced from MIMIC-CXR (Johnson et al., 2019) and Saab et al. (2020) with $n_{\text{text}} = 159,830$ and $n_{\text{text}} = 41,258$, respectively.

## A.5 EXTENDED RESULTS

### A.5.1 EXTENDED ANALYSIS OF CROSS-MODAL EMBEDDINGS

In Section 6.1, we explore the effect of embedding type on slice discovery performance. Here, we provide an extended evaluation of our results.

We note that slice discovery performance is generally lower across rare slices when compared to correlation and noisy label slices. This trend is visible for both model types (synthetic and trained) and all embedding types (unimodal and cross-modal). This is likely due to the nature of the rare slice setting; since a rare subclass occurs in the dataset with very low frequency, it is difficult for SDMs to identify the error slice.

Slice discovery performance on synthetic models is often higher than performance on trained models. This is expected because the use of synthetic models allows us to explicitly control performance on our labeled ground-truth slices, and as a result, Domino can more effectively recover the slice. On the other hand, trained models are likely to include underperforming, coherent slices that are not labeled as "ground-truth", which may limit the ability of Domino to recover the labeled slice. Trained models are also likely to exhibit lower slice performance degradations than synthetic models. We discuss these trade-offs in Section 4.1.2.

### A.5.2 EXTENDED ANALYSIS OF ERROR-AWARE MIXTURE MODEL

In Section 6.2, we explore how the choice of slicing algorithm affects slice discovery performance. Here, we provide an extended evaluation of our results.

Additional experimental results with our error-aware mixture model are shown in Figure 8.

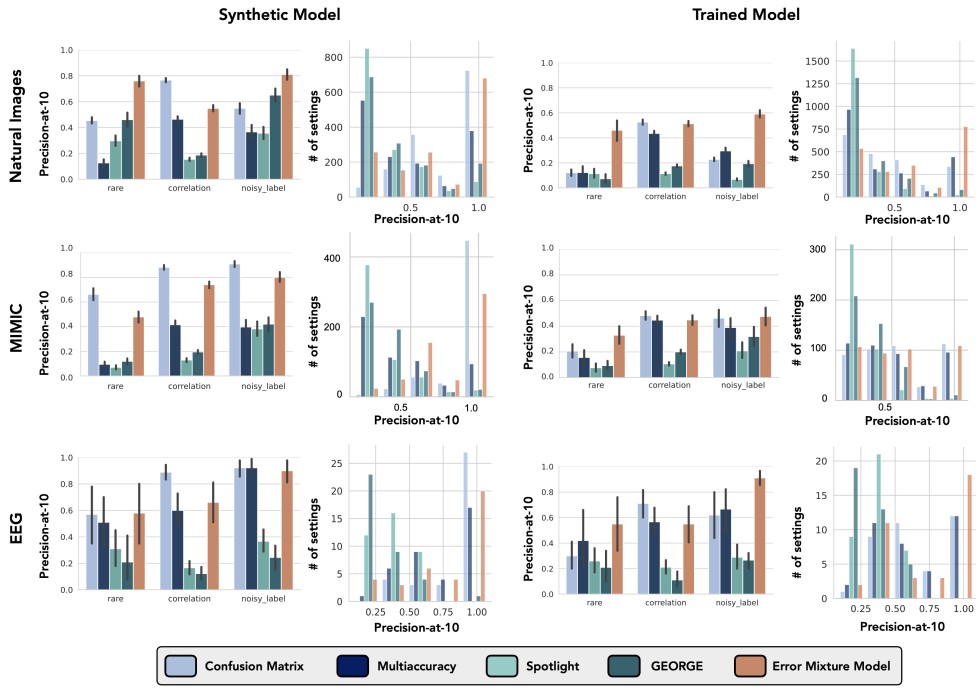

Figure 8: **Error-aware mixture model enables accurate slice discovery**. We show that when cross-modal embeddings are provided as input, our error-aware mixture model often outperforms previously-designed SDMs.

We note that the naive Confusion SDM demonstrates high performance on correlation slices across all three datasets, even outperforming our error-aware mixture model in some cases. This finding suggests that when a strong correlation exists, simply inspecting the confusion matrix may be sufficient for effective slice discovery.

Our error-aware mixture model demonstrates significantly higher performance on rare slices than prior SDMs; this is especially visible in trained model results. This is likely because our error-aware mixture model jointly models the input embeddings, class labels, and model predictions, allowing for better identification of rare slices when compared to existing methods.

### A.5.3 EXTENDED ANALYSIS OF NATURAL LANGUAGE DESCRIPTIONS

Please refer to Figure 9 for a quantitative evaluation of natural image descriptions.

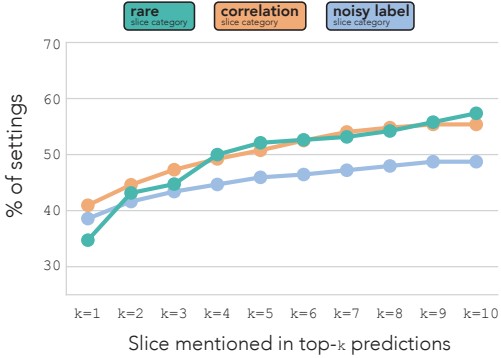

Figure 9: **Descriptions of discovered slices align with the names of the ground truth slices**. Here, we show the fraction of natural image settings where Domino includes the exact name of the ground truth slice (or one of its WordNet synonyms (Fellbaum, 1998)) in the top-$k$ slice descriptions.

### A.5.4 FUTURE WORK

Based on our analysis of model trends, we identify several directions for future work. First, we observe that slice discovery is particularly difficult when the strength of the slice ($\alpha$) is low. In the future, we aim to explore strategies for improving slice discovery and explanation generation in this scenario. Additionally, we hope to explore strategies for generating informative input embeddings when access to paired input-text data is limited. Finally, we intend to run controlled user studies in order to understand when explanations generated by Domino are actionable for practitioners and domain experts.

## A.6 GLOSSARY OF NOTATION

### Classification (Section 3)

$\mathcal{X}$ The set of values that the inputs can take on in a standard classification setting. For example, this could be the set of all possible $256 \times 256$ RGB images.

$\mathcal{Y}$ The set of values that the labels can take on in a standard classification setting. In this work, we deal primarily with binary classification where $\mathcal{Y} = \{0, 1\}$.

$X$ A random variable representing the input in a standard classification setting.

$Y$ A random variable representing the label in a standard classification setting.

$P$ A probability distribution. For example, the joint distribution over inputs and labels can be expressed as $P(X, Y)$.

$x_i$ The realization of the $i^{\text{th}}$ sample of $X$.

$y_i$ The realization of the $i^{\text{th}}$ sample of $Y$.

$n$ The number of samples in a classification dataset.

$\mathbf{x}$ The set of $n$ samples of $X$, such that $\mathbf{x} = \{x_i\}_{i=1}^n \in \mathcal{X}^n$.

$\mathbf{y}$ The set of $n$ samples of $Y$, such that $\mathbf{y} = \{y_i\}_{i=1}^n \in \mathcal{Y}^n$.

$\mathcal{D}$ A labeled dataset sampled from $P(X, Y)$, such that $\mathcal{D} = \{(x_i, y_i)\}_{i=1}^n$

$h_\theta$ A classifier with parameters $\theta$ that predicts $Y$ from $X$, where $h_\theta : \mathcal{X} \to \mathcal{Y}$.

$\hat{Y}$ A random variable representing the prediction of the model, such that $\hat{Y} = h_\theta(X)$.

$\ell$ A performance metric for a standard classification setting, where $\ell : \mathcal{Y} \times \mathcal{Y} \to \mathbb{R}$ (*e.g.* accuracy).

### Slice Discovery (Section 3)

$S^{(j)}$ A random variable representing the $j^{\text{th}}$ data slice. This is a binary variable $S^{(j)} \in \{0, 1\}$.

$s_i^{(j)}$ The realization of the $i^{\text{th}}$ sample of $S^{(j)}$.

$k$ The number of slices in the data.

$\mathbf{S}$ A random variable representing a set of $k$ slices $\mathbf{S} = \{S^{(j)}\}_{j=1}^k \in \{0, 1\}^k$.

$\mathbf{s}_i$ The realization of the $i^{\text{th}}$ sample of $\mathbf{S}$.

$\psi^{(j)}$ A slicing function $\psi^{(j)} : \mathcal{X} \times \mathcal{Y} \to \{0, 1\}$

$\hat{k}$ The number of slicing functions returned by a slice discovery method.

$\Psi$ A set of $\hat{k}$ slicing functions $\Psi = \{\psi^{(j)}\}_{j=1}^{\hat{k}}$.

$M$ A slice discovery method, $M(\mathcal{D}, h_\theta) \to \Psi$.

### Evaluation Framework (Section 4)

$L$ A slice discovery metric, $L : 0, 1^k \times [0, 1] \to \mathbb{R}$ (*e.g.* precision-at-10).

$C$ A random variable representing a metadata attribute. We use $C$ to generate datasets with underperforming slices.

$\alpha$ The strength of a generated slice. For example, in a correlation slice, $\alpha$ is the Pearson correlation coefficient between the label $Y$ and the correlate $C$.

$\epsilon$ The performance degradation in metric $\ell$ required for a model to be considered underperforming.

$\bar{h}_\theta$ A synthetic model, $\bar{h} : [0, 1]^k \times \mathcal{Y} \to [0, 1]$, which samples predictions $\hat{Y}$ conditional on $Y$ and $\mathbf{S}$.

**Domino** (Section 5)

| | |
|---|---|
| $\mathcal{T}$ | The set of possible text strings. |
| $T$ | A random variable representing a text string in a paired data setting, $T \in \mathcal{T}$. |
| $V$ | A random variable representing inputs in a paired data setting, $V \in \mathcal{X}$. Note that we do not use $X$ here in order to emphasize the difference between the data used to train the classifier and the data used to learn cross-modal embeddings. |
| $n_{\text{paired}}$ | The number of examples in a paired dataset. |
| $\mathcal{D}_{\text{paired}}$ | A paired dataset $\mathcal{D} = \{(v_i, t_i)\}_{i=1}^{n_{\text{paired}}}$, where the text $t_i$ describes the input $v_i$. |
| $d$ | The dimensionality of the embeddings. |
| $g_{\text{input}}$ | An embedding function for inputs, $g_{\text{input}} : \mathcal{X} \to \mathbb{R}^d$. |
| $g_{\text{input}}$ | An embedding function for text, $g_{\text{text}} : \mathcal{T} \to \mathbb{R}^d$. |
| $Z$ | A random variable representing the embedding of an input, such that $Z = g_{\text{input}}(X)$. |
| $z_i$ | The value of the $i^{\text{th}}$ sample of $Z$, such that $z_i = g_{\text{input}}(x_i)$. |
| $\mathbf{z}$ | The set of $n$ samples of $Z$, such that $\mathbf{z} = \{z_i\}_{i=1}^n \in \mathbb{R}^{n \times d}$. |
| $Z_{\text{text}}$ | A random variable representing the embedding of a text string, such that $Z = g_{\text{text}}(T)$. |
| $z_i^{\text{text}}$ | The realization of the $i^{\text{th}}$ sample of $Z_{\text{text}}$, such that $z_i^{\text{text}} = g_{\text{text}}(t_i)$. |
| $\bar{z}_{\text{slice}}^{(i)}$ | The average embedding of the $i^{\text{th}}$ slice. |
| $\bar{z}_{\text{class}}^{(i)}$ | The average embedding of the $i^{\text{th}}$ class. |
| $\mu^{(i)}$ | In the error-aware mixture model, the mean parameter of the Gaussian distribution used to model $Z$ for the $i^{\text{th}}$ slice. |
| $\mathbf{\Sigma}^{(i)}$ | In the error-aware mixture model, the covariance parameter of the Gaussian distribution used to model $Z$ for the $i^{\text{th}}$ slice. |
| $\mathbf{p}^{(i)}$ | In the error-aware mixture model, the parameter of the categorical distribution used to model $Y$ for the $i^{\text{th}}$ slice. |
| $\hat{\mathbf{p}}^{(i)}$ | In the error-aware mixture model, the parameter of the categorical distribution used to model $\hat{Y}$ for the $i^{\text{th}}$ slice. |
| $\phi$ | The parameters of the error aware mixture model: $\phi = [\mathbf{p}_S, \{\mu^{(s)}, \Sigma^{(s)}, \mathbf{p}^{(s)}, \hat{\mathbf{p}}^{(s)}\}_{s=1}^{\bar{k}}]$ |

