# OpenReview forum: "Domino: Discovering Systematic Errors with Cross-Modal Embeddings"
_ICLR.cc/2022/Conference — ICLR 2022 Oral_

### Official Review · Reviewer_hQrb · 2021-10-31

**Correctness:** 2
**Technical Novelty And Significance:** 2
**Empirical Novelty And Significance:** 2
**Recommendation:** 6
**Confidence:** 3

**Main Review:**

General comments:

The paper addresses the problem of identifying on which subsets of data machine learning models make systematic errors. The authors term this as a "slice discovery method" (SDM). Two desirable properties for a SDM are outlined: First, providing a quantitative evaluation framework for measuring performance of SDMs and secondly, ensuring a solution that is "coherent". Here coherence is defined as being understandable by a domain experts, The proposed evaluation framework seems rather ad-hoc and poorly justified. The main idea, the DOMINO approach seems limited to images with captions. In the approach images and captions are separately embedded while preserving their similarity. This is followed by a mixture model which identifies errors in the model predictions, but it is not clear to me where the actual model predictions are trained. Also, the paper is often written in a confusing manner that makes it difficult to follow and understand the the contributions of the paper. Furthermore there is a lack of definitions for many of the terminology used in the paper.

Specific comments:

- I find the term "slice discovery method" misleading, it is not commonly used term in this field.
- The definition of the slice discovery problem (section 2) is rather imprecise and uses formulations such as "exhibits degraded performance". No precise definition is given what "degraded" means in this context.
- After reading the definition of the slice discovery problem (section 2), I am still not clear whether a slice can refer to whole images only or part of the images (or frames in a video).


**Summary Of The Paper:**

The paper propose a framework for identifying on which subsets of data machine learning models make systematic errors. The problem is cast in two parts: (1) identify a model that can be identify a subset of data and predict degraded performance of the machine learning model for this subset and (2) ensure that the identified subset is "coherent". The framework is evaluated on a number of classification tasks in computer vision and medicine.

**Summary Of The Review:**

The paper addresses an interesting problem but is hampered by the ad-hoc nature of the approach and the lack of clarity in the problem formulation and writing.

---

> ### Author Response · Authors · 2021-11-19
> **Response to Reviewer hQrb (1/2)**
>
> We thank Reviewer HQRB for reviewing our paper and providing helpful feedback on our work. We address many of Reviewer HQRB’s concerns in the general response above, and we provide additional details on specific comments below.
>
> **Justification of the evaluation framework.**
> > *“The proposed evaluation framework seems rather ad-hoc and poorly justified.”*
>
> Our evaluation framework enables the first large-scale, quantitative evaluation of slice discovery methods (SDMs). Prior works (see references [1,2,3]) either use (1) qualitative evaluations (*i.e.* subjective interpretations of discovered slices) or (2) include fewer than ten slice discovery settings, all from a single slice category. In our work, we propose a framework for programmatically generating a large number of realistic slice discovery settings. We use our framework to instantiate over 1,000 settings across diverse slice types and application domains, which we will release publicly.
>
> The design of our evaluation framework was motivated by an extensive review of real-world slices documented in prior works (this literature review is included in Section A.1 of the revised submission). Below we provide justification for each aspect of our evaluation framework, and provide pointers to the sections in the manuscript where these design choices are discussed:
>
> - **Slice Categories.** Our framework includes three slice categories: rare slices, correlation slices, and noisy label slices. All of the slices that we found in our literature review fall into one of these three categories (see Section A.1). Prior work on hidden stratification highlights a similar set of slice categories [4]. Other slice categories could easily be incorporated into the framework in future work. Additionally, within each category, we generate slices with varying strengths in order to reflect the diversity of slices likely to occur in real-world datasets. This is discussed in further detail in Section 4.1.1.
> - **Metrics.** In order to evaluate the quality of a predicted slice, practitioners typically analyze the top-$k$ examples in the slice [1,2,3]. Our primary metric precision-at-$k$ aligns closely with this intended use case.
> - **Application domains.** To capture the diversity of machine learning tasks, we generate slices in three diverse domains: natural images (CelebA and ImageNet), medical images (MIMIC), and medical time-series data (EEG). Since our evaluation framework is programmable, it can be easily expanded to cover additional domains. This is discussed in further detail in Section 4.2.
>
> If there remain aspects of the framework that feel poorly justified to Reviewer HQRD, we would appreciate additional feedback.
>
> **Clarification.**
> > *"The Domino approach seems limited to images with captions."*
>
> Domino is not limited to images with captions. In our experiments, we apply Domino to ImageNet and CelebA, two datasets without captions. Domino does use *pretrained* cross-modal embeddings trained on available corpora of paired data, but it does not require the slice discovery dataset $\mathcal{D}$ to be captioned. In many domains (*e.g.* natural images and medical images), cross-modal embeddings are readily available for download [8,9], so the average Domino user would not need access to any captioned data. In more specialized domains where pretrained cross-modal embeddings are unavailable (such as non-image inputs), paired input-text data may still be abundantly available, allowing the practitioner to train a domain-specific set of cross-modal embeddings. In our work, we utilize this procedure for our medical time-series (EEG) dataset, where we train a set of custom cross-modal embeddings that align representations of time-series signals with text from physician reports.
>
> **Paper Organization.**
> > *"The paper is written in a confusing manner.”*
>
> We thank Reviewer HQRB for the suggestion to improve paper organization. We have modified the manuscript and refer the reviewer to the general response above, where we detail the structural changes that were made.

---

> > ### Author Response · Authors · 2021-11-19
> > **Response to Reviewer hQrb (2/2)**
> >
> > **Terms and Definitions.**
> > > *There is a lack of definitions for many of the terminology... I find the term ‘slice discovery method’ misleading, it is not commonly used term in the field... The definition of the slice discovery problem (Section 2) is rather imprecise...”*
> >
> > We have modified Sections 1, 2, 3, and 4 to provide additional details about the slice discovery problem setting. Specifically, we have made the following changes:
> >
> > - We have added a definition of the term *slice* to the Introduction.
> > - We have expanded the Related Work section to provide additional context on prior slice discovery approaches.
> > - We have edited Section 3 to provide a clearer description of the slice discovery problem setting.
> > - The exact definition of “degraded performance” will vary by problem setting. In our experiments, we define degraded performance as a 10% drop in classification performance across examples in the slice when compared to examples outside the slice. This has been updated in Section 4.1.2.
> >
> > Since most slice discovery methods have been proposed only recently, prior work uses a number of terms to refer to such methods, and there is no consensus on a name. Slice discovery methods have been referred to as “slice finders” [5],  “methods for discovering systematic errors” [1], “multiaccuracy auditors” [3] and “subclass label estimation” [2]. We use the term “slice discovery” because it is concise and intuitive, and it reconciles the diverse names found in the literature. We note that the term “slice” is standard and has been used in several recent works to refer to data subgroups united by a shared characteristic [6,7]. We would appreciate a comment if there are specific ways in which Reviewer HQRB found the name misleading.
> >
> > **Clarification.**
> > > *“It is not clear to me where the actual model predictions are trained...No precise definition is given what "degraded" means in this context”*
> >
> > We describe the model training procedure in Section 4.1.2. The implementation of $h_\theta$ will depend on the application setting. For our natural image and medical image settings, $h_\theta$ takes the form of a ResNet-18 model trained across the provided dataset $D$. For our medical time-series data, predictions are generated using a Dense Inception CNN. We ensure that models exhibit degraded performance across the specified slice, which we define as at least a 10% drop in classification performance across examples in the slice when compared to examples outside the slice (the performance metric we use is either recall or AUROC depending on the dataset and task).
> >
> > **Clarification**
> > > *"After reading the definition of the slice discovery problem (Section 2), I am still not clear whether a slice can refer to whole images only or part of the images (or frames in a video)."*
> >
> > The term *slice* refers to a group of data examples united by a shared attribute (see section 3). For example, if we consider the CelebA dataset which includes images of celebrity faces, some examples of data slices include images with blonde hair, individuals with beards, and individuals with earrings. We have clarified the definition of the term *slice* in the updated manuscript in Section 1.

---

> > ### Public Comment · ~Claude_Ross1 · 2024-12-23
> > **Thanks**
> >
> > We appreciate Reviewer HQRB's detailed feedback and have clarified the evaluation framework, Domino's applicability, and improved paper organization based on suggestions https://thatsnotmy-neighbor.io to enhance clarity and justification.

---

> ### Author Response · Authors · 2021-11-22
> **Dear reviewer hQrb: we'd love to know if you have any more questions after our response**
>
> Dear reviewer hQrb
>
> Thank you very much for your helpful feedback and suggestions, they helped us to improve the paper. We tried to carefully address all of your comments in our response and the updated paper. Please let us know if you have any further questions, and we are very happy to follow up!
>
> Thank you for your time!

---

### Official Review · Reviewer_LuSN · 2021-11-02

**Correctness:** 3
**Technical Novelty And Significance:** 2
**Empirical Novelty And Significance:** 3
**Recommendation:** 8
**Confidence:** 2

**Main Review:**

The paper addresses interesting problems in the area of slice discovery, particularly underperforming clusters. Overall the Domino approach seems to work well for this task. There are ablation studies on the embeddings used and clustering algorithm chosen. The part about generating natural language explanation is not as convincing as it's ultimately using the text embeddings on a large corpus in a retrieval setting. Actually generating text from the given slice embeddings would have been more convincing. Currently the text seems to be restricted to single words from Fig 5.
Though the idea introduced in the paper is quite interesting, the paper itself is organised in a very confusing manner. Related work is only introduced in section 6 on page 9 and seems incomplete. The Domino method is shown in Figure 1 on page 2 but the actual text describing it is on page 6. Another weakness is the novelty. Though the framework is novel, the individual models are not. Given the new evaluation framework, it would have been great to introduce some further technical novelty.
In the experiments section, the results are described as-is without further interpretation. Do the results with the evaluation framework indicate specific trends? If so, why? Maybe further investigation on this would provide some ideas for future exploration - which are also missing in the paper.

**Summary Of The Paper:**

The paper introduces a new approach for slice discovery by leveraging the advances in cross-modal embeddings. The paper also suggests a new evaluation framework to quantitively evaluate SDMs.

**Summary Of The Review:**

The paper introduces novel frameworks to evaluate and perform slice discovery. Though technical novelty in the individual parts are limited, the overall framework seem to be interesting and perform well according to the experimental results. The understanding of the paper suffers from the way how it is organised and should be further improved.

---

> ### Author Response · Authors · 2021-11-19
> **Response to Reviewer LuSN (1/2)**
>
> We thank Reviewer LuSN for their positive comments and for providing thoughtful feedback on our work. We address many of Reviewer LuSN’s comments in our general response above, and we provide additional details on specific comments below.
>
> **Paper Organization.**
> >*"Though the idea introduced in the paper is quite interesting, the paper itself is organised in a very confusing manner. Related work is only introduced in section 6 on page 9 and seems incomplete. The Domino method is shown in Figure 1 on page 2 but the actual text describing it is on page 6."*
>
> We appreciate the reviewer’s thoughtful suggestions for improving the organization of our paper. We refer Reviewer LuSN to our general response where we describe a significant reorganization of the paper.
>
> **Technical Novelty.**
> > *"Though the framework is novel, the individual models are not. Given the new evaluation framework, it would have been great to introduce some further technical novelty.”*
>
> We thank Reviewer LUSN for noting the novelty of our evaluation framework. We clarify that our manuscript does introduce a novel technical method for performing slice discovery. Specifically, we introduce Domino, which is an SDM with the following novel technical contributions:
>
> 1. ***Embed:*** Motivated by the insight that cross-modal embeddings produce semantically meaningful representations, we leverage cross-modal embeddings to discover coherent slices. Our work is the first to use cross-modal embeddings for slice discovery and demonstrate empirically that they outperform uni-modal embeddings at this task. Additionally, while cross-modal embeddings have been trained for natural images and medical images, to the best of our knowledge our work is the first to train cross-modal embeddings for continuous medical time-series (EEG) data.
> 2. ***Slice:*** We propose a novel error-aware mixture model that jointly models the input embeddings, class labels, and model predictions to find slices in embedding space. We demonstrate empirically that it outperforms existing methods by 12 percentage points via detailed ablation studies. To the best of our knowledge, such a mixture model has not previously been proposed.
> 3. ***Describe:*** Our approach to generating natural language descriptions for discovered slices is novel. Prior work has not introduced any other methods for describing discovered slices with natural language.
>
> To summarize, in addition to our evaluation framework, we propose a slice discovery method that leverages multiple novel components to effectively identify and describe slices.
>
> **Framing language generation as a retrieval task.**
> > *"The part about generating natural language explanation is not as convincing as it's ultimately using the text embeddings on a large corpus in a retrieval setting...Currently the text seems to be restricted to single words from Fig 5.”*
>
> Although the version of Figure 5 included in our original submission included text descriptions consisting of only one word, our approach can also be used to generate full-sentence descriptions, which we demonstrate in the updated manuscript with a set of new experiments. To do so, we apply pre-trained masked-language models to domain-specific templates in order to generate a corpus of candidate full-sentence descriptions. For example, when using the CelebA dataset, we use pre-trained BERT to generate thousands of candidate descriptions by filling in templates of the form “A photo of a person [MASK] [MASK] [MASK].”  We opted for this approach rather than unconstrained generation because it affords the practitioner greater control over the generation process due to the predefined templates. In our updated manuscript, we have added Section A.3.2, which describes this approach in detail.
>
> For our medical applications, we also generate full-sentence natural language descriptions using a corpus of complete physician reports. Descriptions accurately characterize identified slices. Examples are displayed in Section A.4.3.
>
> We agree with the reviewer that exploring the feasibility of unconstrained generation from the embeddings would be an interesting direction to explore in future work.

---

> > ### Author Response · Authors · 2021-11-19
> > **Response to Reviewer LuSN (2/2)**
> >
> > **Discussion and future work.**
> > > *“In the experiments section, the results are described as-is without further interpretation. Do the results with the evaluation framework indicate specific trends? If so, why? Maybe further investigation on this would provide some ideas for future exploration - which are also missing in the paper.”*
> >
> > We thank the reviewer for raising this point.  Below, we include additional interpretation of trends from our experiments. We provide a detailed analysis of trends in Appendix Section A.4 in our updated manuscript, and we have updated the conclusion to include some ideas for future exploration.
> >
> > In Section 6.1, we explore the effect of embedding type on slice discovery performance:
> >
> > 1. We find that uni-modal embeddings (BiT, ImageNet, trained activations of $h_\theta$) are particularly effective for correlation slices, exhibiting only a small decrease in performance when compared to cross-modal embeddings. This makes sense because a model that relies on a spurious correlate to make predictions will likely capture information about the correlate in its activations.
> > 2. We note that slice discovery performance is generally lower across rare slices when compared to correlation and noisy label slices. This trend is visible for both model types (synthetic and trained) and all embedding types (unimodal and cross-modal). This is likely due to the nature of the rare slice setting; since a rare subclass occurs in the dataset with very low frequency, it is difficult for SDMs to identify the error slice.
> > 3. Slice discovery performance on synthetic models is often higher than performance on trained models. This is expected because the use of synthetic models allows us to explicitly control performance on our labeled ground-truth slices, and as a result, Domino can more effectively recover the slice. On the other hand, trained models are likely to include underperforming, coherent slices that are not labeled as “ground-truth”, which may limit the ability of Domino to recover the labeled slice. Trained models are also likely to exhibit lower slice performance degradations than synthetic models. We discuss these trade-offs in Section 4.1.2.
> >
> > In Section 6.2, we explore the effect of slicing algorithms on performance:
> >
> > 1. We note that the naive Confusion SDM demonstrates high performance on correlation slices across all three datasets, even outperforming our error-aware mixture model in some cases. This finding suggests that when a strong correlation exists, simply inspecting the confusion matrix may be sufficient for effective slice discovery.
> > 2. Our error-aware mixture model demonstrates significantly higher performance on rare slices than prior SDMs; this is especially visible in trained model results. This is likely because our error-aware mixture model jointly models the input embeddings, class labels, and model predictions, allowing for better identification of rare slices when compared to existing methods.
> >
> > We also outline some directions for future work:
> >
> > - Slice discovery is particularly difficult when the strength of the slice ($\alpha$) is low. In the future, we aim to explore strategies for improving slice discovery and explanation generation in this scenario.
> > - We hope to explore strategies for generating informative input embeddings when access to paired input-text data is limited.
> > - In our response to Reviewer si46, we provide a preliminary user evaluation in order to determine if generated slice descriptions are actionable. We plan to expand this in the future with a controlled user study in order to better understand when explanations are actionable.

---

### Official Review · Reviewer_si46 · 2021-11-02

**Correctness:** 4
**Technical Novelty And Significance:** 3
**Empirical Novelty And Significance:** 3
**Recommendation:** 8
**Confidence:** 2

**Main Review:**

Pros:

I believe that there is well backed motivation for work based off of the plentiful literature review.
There is novel integration of CLIP as well as SDMs previously not combined before
A variety of datasets seem to show that the proposed method is useful.
Code is available for reproducibility.

Cons:

I would rather the the literature review section be in the intro rather than in the conclusion for flow. Then, the authors could summarize their work in the conclusion instead.

Are textual descriptions of the Slices actually actionable for real life experts? I would like if physicians found the textual descriptions of the MIMIC and EEG dataset slices useful.


**Summary Of The Paper:**

Recent studies have proposed automated slice discovery methods (SDMs), which leverage learned model representations to mine input data for slices, or important subgroups of data, on which a model performs poorly.

An ideal SDM should automatically identify:
1. Slices that contain examples on which the model underperforms, or has a high error rate.
2. Slices that contain examples that are coherent, or align closely with a human-understandable concept.

This is difficult because:
1: No quantitative evaluation framework exists for measuring performance of SDMs; Existing SDM evaluations are either qualitative, performed on synthetic data, or consider only a small subset.
2. Prior qualitative evaluations have demonstrated that existing SDMs often identify slices that are incoherent, even though they may satisfy the first ideal case.

Domino:
The authors preset Domino, an SDM that leverages cross-modal embeddings and a novel error-aware mixture model to discover and describe coherent slices using natural language descriptions. The proposed method could also quantitatively compare SDMs, which has not been done before.

Step 1. Embed: Encode inputs in a cross-modal embedding space with a function.

Step 2. Slice:  identify underperforming regions in the cross-modal embedding space using an error-aware mixture model fit on the input embeddings, model predictions, true class labels using expectation maximization. I.e. input embeddings, class labels, and model predictions as independent based on slice.

Step 3. Describe: Use the text embedding function ψtext learned in step (1) to generate a set of k natural language descriptions of the discovered slice.

Evaluation Approach of 3 popular slide types:
Rare slice: To generate settings with rare slices, Construct a skewed dataset such that for a given class label Y, elements in subclass C occur with proportion α, where 0.01 < α < 0.1.
Correlation slice: Construct a dataset such that a linear correlation α exists between the target variable and other class labels, where 0.2 < α < 0.8.
Noisy label slice. Construct dataset such that for each given class label Y, the elements in subclass C exhibit label noise with probability α, where 0.01 < α < 0.3.

Experiments show that when cross-modal embeddings are provided as input, the error-aware mixture model often outperforms previously-designed SDMs.

**Summary Of The Review:**

I would tend to accept this paper as it is novel enough and supported by empirical experiments.

---

> ### Author Response · Authors · 2021-11-19
> **Response to Reviewer si46**
>
> We thank Reviewer si46 for positive comments and helpful feedback on our work. We address many of Reviewer si46’s suggestions in the general response above and respond to specific comments below.
>
> **Paper Organization.**
> >*"I would rather the literature review section be in the intro rather than in the conclusion for flow.”*
>
> We thank Reviewer si46 for the suggestion to move the literature review section and expand the conclusion. We have moved the literature review to Section 2 and modified the manuscript. We refer the reviewer to the general response above, where we detail the structural changes that were made.
>
> **Actionable Natural Language Descriptions.**
> > “*Are textual descriptions of the slices actually actionable for real life experts?”*
>
> We thank Reviewer si46 for this question. We believe a thorough user study where we provide textual descriptions to experts for interpretation would be a promising direction for future work. Although conducting such an analysis is out of scope for this study, we provide a preliminary version below that seems promising:
>
> We displayed the top 10 EEG slice descriptions generated by Domino to a board-certified neurologist who regularly administers EEGs. We selected a correlation slice setting, where the presence of seizures (target variable) was strongly correlated (alpha=0.8) with young patients less than 1 year in age (slice). We asked the neurologist two questions: (1) can you identify a coherent concept that is shared among the 10 descriptions? (2) What actions would you take after being aware of this systematic error?
>
> 1. *Finding a coherent concept*: The neurologist explained that the sentences are mainly describing patients that previously had, currently have, or are likely to develop seizures, where a specific seizure type (focal) was more prominent. He also found that the majority of the descriptions make it clear that the patients are young.
> 2. *Taking actions*. The neurologist explained that they would expect the bias caused by age to be reflected in how the model interprets the EEG signal, since younger patients exhibit higher amplitudes and slower waveforms than adults. Moreover, the neurologist would expect the model to perform differently on specific seizure types since older patients have more focal seizures, while pediatric patients might have more myoclonic, migraine, and genetic seizures. As a result, the neurologist would take action to further evaluate model performance on seizure subtypes and sample more EEGs from older patients.
>
> In the future, we hope to conduct controlled user studies to directly test the real-world effectiveness of Domino. We believe a comprehensive, well-executed user study to be out of scope for this paper.
>
> We again thank Reviewer si46 for their review of our manuscript, and we hope that the above responses adequately address all concerns.

---

> > ### Comment · Reviewer_si46 · 2021-11-29
> > **Thank you for your response**
> >
> > I now feel more strongly towards accept. Best of luck!

---

### Author Response · Authors · 2021-11-19
**Response to all reviewers (1/2)**

We thank the reviewers for their thoughtful and constructive review of our manuscript. We were encouraged to hear the reviewers found the slice discovery problem we present to be both interesting (Reviewers LuSN, hQrb) and well-motivated (Reviewer si46) and that they view our methodology as useful (Reviewer si46), novel (Reviewer si46), and effective (Reviewers si46, LuSN). In response to feedback, we provide a general response here to points raised by multiple reviewers, individual responses below to address each reviewer’s concerns, and an updated manuscript.

Regarding questions from Reviewers LuSN and hQrb about **novelty and technical contributions**, we emphasize that the two key contributions of this work are (1) designing a novel quantitative evaluation framework for rigorously evaluating slice discovery methods (SDMs), which we instantiated with over 1,000 diverse slice settings, and (2) introducing *Domino*, an SDM that leverages cross-modal embeddings and a novel error-aware mixture model to both identify slices and provide natural language explanations.

1. Prior works [1,2,3] have proposed slice discovery methods for unstructured inputs, and these approaches are evaluated either qualitatively or with a small selection of tasks and slices. To the best of our knowledge, our paper is the first to introduce a large-scale quantitative evaluation framework for rigorously evaluating SDMs.
2. Prior approaches often extract incoherent slices, where slices are not unified by a single, actionable concept. We address this by utilizing cross-modal representations (i.e. input embeddings that are aligned with textual phrases), which have not been previously used for slice discovery. Our work is also the first to generate natural language explanations for slices. We demonstrate through experiments with our evaluation framework that Domino often outperforms prior SDMs across a range of classification tasks, application domains, and slice categories.

As summarized by Reviewer si46, we perform a first-of-its-kind comparison of existing SDMs and introduce Domino, an SDM powered by a novel application of cross-modal embeddings and a novel error-aware mixture model.

In response to comments from Reviewers si46 and LuSN about **natural language descriptions**, we have updated our manuscript to include a larger, more diverse set of explanations for our natural image, medical image, and medical time-series datasets. As demonstrated in Section A.4.3, our approach can successfully generate explanations for a range of slice types and slice settings. We also ran a set of additional experiments in order to demonstrate that full-sentence descriptions can be generated with our method. We include details on these new experiments in Sections A.3.2 and A.4.3 of the revised manuscript.

Regarding feedback from Reviewers LuSN, si46, and hQrb about the **organization** of our manuscript, we have made the following changes.

- Introduction (Section 1): We have added definitions for some terminology and clarified our problem setting.
- Figure 1 (Section 1): Figure 1 provides an overview of our two key contributions: (1) the evaluation framework, and (2) Domino, our novel SDM. We have edited the figure and caption to better communicate our contributions.
- Related Work (Section 2): We have significantly expanded our literature review section to provide additional background on slices, the slice discovery task, and cross-modal embeddings, and we have moved the related work to Section 2 as suggested.
- Slice Discovery Preliminaries (Section 3): We have modified the formal definition of the slice discovery task in order to improve clarity.
- Slice Discovery Evaluation Framework (Section 4): We have added additional details to the slice category descriptions in 4.1.1 and added further justification for our evaluation framework.
- Domino (Section 5): We have updated section 5.1 to clarify assumptions under which paired input-text data is necessary. We have also updated section 5.3 and added Appendix Sections A.3.2 and A.4.3 to provide details on generating sentence-level explanations.
- Experiments (Section 6): We have added additional information on experiments, observed trends, and explanations in Appendix Section A.4.
- Conclusion (Section 7): We have updated the conclusion to include directions for future work. We also provide an extended discussion on future work in Appendix Section A.4.4.
- Appendix: We have updated the appendix to include a survey of slices in the wild (A.1), additional details on generating natural language descriptions (A.3.2), extended evaluation on the trends observed in Section 6 (A.4.1 and A.4.2), an extended analysis of the natural language descriptions generated by Domino (A.4.3), and possible directions for future work (A.4.4).

---

> ### Author Response · Authors · 2021-11-19
> **Response to all reviewers (2/2)**
>
> In summary, as machine learning models become increasingly widespread in use, slice discovery will become critical for building robust models, especially in safety-critical settings where the presence of hidden underperforming slices could result in significant consequences. We hope that our evaluation framework will facilitate the development of effective systems for this task and that future work will build on our proposed method, Domino.
>
> We would again like to thank all reviewers for their time and feedback, and we hope that our changes adequately address all concerns.
>
> **References**
>
> [1] Greg d’Eon, Jason d’Eon, James R. Wright, and Kevin Leyton-Brown. The spotlight: A general method for discovering systematic errors in deep learning models, 2021.
>
> [2] Nimit Sohoni, Jared Dunnmon, Geoffrey Angus, Albert Gu, and Christopher Re. No sub- ´ class left behind: Fine-grained robustness in coarse-grained classification problems. In H. Larochelle, M. Ranzato, R. Hadsell, M. F. Balcan, and H. Lin (eds.), Advances in Neural Information Processing Systems, volume 33, pp. 19339–19352. Curran Associates, Inc., 2020.
>
> [3] Michael P Kim, Amirata Ghorbani, and James Zou. Multiaccuracy: Black-Box Post-Processing for Fairness in Classification. arXiv:1805. 12317 [cs, stat], August 2018.
>
> [4] Luke Oakden-Rayner, Jared Dunnmon, Gustavo Carneiro, and Christopher Re. Hidden stratification causes clinically meaningful failures in machine learning for medical imaging. September 2019.
>
> [5] Chung, Yeounoh, Tim Kraska, Neoklis Polyzotis, Ki Hyun Tae, and Steven Euijong Whang. 2019. “Slice Finder: Automated Data Slicing for Model Validation.” In *2019 IEEE 35th International Conference on Data Engineering (ICDE)*, 1550–53. Ieeexplore.ieee.org.
>
> [6] Karan Goel, Nazneen Rajani, Jesse Vig, Samson Tan, Jason Wu, Stephan Zheng, Caiming Xiong, Mohit Bansal, and Christopher Re. Robustness Gym: Unifying the NLP Evaluation Landscape. ´ arXiv:2101. 04840 [cs], January 2021
>
> [7] Mayee Chen, Karan Goel, Nimit Sohoni, Fait Poms, Kayvon Fatahalian, Christopher Re. Mandoline: Model Evaluation under Distribution Shift. *International Conference on Machine Learning*, 2021.
>
> [8] Alec  Radford,  Jong  Wook  Kim,  Chris  Hallacy,  Aditya  Ramesh,  Gabriel  Goh,  Sandhini  Agarwal,  Girish  Sastry,  Amanda  Askell,  Pamela  Mishkin,  Jack  Clark,  Gretchen  Krueger,  and  Ilya Sutskever.   Learning  transferable  visual  models  from  natural  language  supervision.   February 2021.
>
> [9] Yuhao  Zhang,  Hang  Jiang,  Yasuhide  Miura,  Christopher  D.  Manning,  and  Curtis  P.  Langlotz. Contrastive  learning  of  medical  visual  representations  from  paired  images  and  text. 2020.

---

### Comment · Reviewer_hQrb · 2021-11-29
**Updated scores**

I would like to thank the authors for addressing most of my concerns. I have adjusted my scores.

---

### Decision · Program_Chairs · 2022-01-20

**Decision:**

Accept (Oral)

**Comment:**

Three experts reviewed this paper and all recommended acceptance. The reviewers liked that the work addressed a common problem in prior related work that it is hard to quantitatively evaluate slide discovery methods. Moreover, the proposed method achieves superior performance over prior arts. Based on the reviewers' feedback, the decision is to recommend the paper for acceptance. The reviewers did raise some valuable concerns, such as paper clarity, significance of the textual descriptions, that should be addressed in the final camera-ready version of the paper. The authors are encouraged to make the necessary changes to the best of their ability. We congratulate the authors on the acceptance of their paper!